# On the Importance of Language-driven Representation Learning for Heterogeneous Federated Learning

**Yunlu Yan**[1*]**, Chun-Mei Feng**[2*]**Wangmeng Zuo**[3,4]**, Salman Khan**[5,6]**, Yong Liu**[2]**, Lei Zhu**[1,7†]

[1] The Hong Kong University of Science and Technology (Guangzhou), China
[2] Institute of High Performance Computing (IHPC),
Agency for Science, Technology and Research (A*STAR), Singapore
[3] Harbin Institute of Technology, China
[4] Pengcheng Laboratory, China
[5] Mohamed bin Zayed University of Artificial Intelligence (MBZUAI), UAE
[6] Australian National University, Australia
[7] The Hong Kong University of Science and Technology, China

## Abstract

Non-Independent and Identically Distributed (Non-IID) training data significantly challenge federated learning (FL), impairing the performance of the global model in distributed frameworks. Inspired by the superior performance and generalizability of language-driven representation learning in centralized settings, we explore its potential to enhance FL for handling non-IID data. In specific, this paper introduces FedGLCL, a novel language-driven FL framework for image-text learning that uniquely integrates global language and local image features through contrastive learning, offering a new approach to tackle non-IID data in FL. FedGLCL redefines FL by avoiding separate local training models for each client. Instead, it uses contrastive learning to harmonize local image features with global textual data, enabling uniform feature learning across different local models. The utilization of a pre-trained text encoder in FedGLCL serves a dual purpose: it not only reduces the variance in local feature representations within FL by providing a stable and rich language context but also aids in mitigating overfitting, particularly to majority classes, by leveraging broad linguistic knowledge. Extensive experiments show that FedGLCL significantly outperforms state-of-the-art FL algorithms across different non-IID scenarios. Codes are available at https://github.com/IAMJackYan/FedGLCL.

## 1 Introduction

Federated learning (Li et al., 2020a) (FL) serves as a powerful tool for distributed learning, allowing multiple clients to collaboratively train a global model without privacy concerns. It is widely applied in various applications (Niknam et al., 2020; Kang et al., 2020), especially in some privacy-sensitive fields such as healthcare (Jiang et al., 2022; Feng et al., 2022; Wicaksana et al., 2022; Yan et al., 2024a; Feng et al., 2023a; Yan et al., 2024b; 2023). The pioneering FL algorithm, FedAvg (McMahan et al., 2017), trains the model locally and averages the updated parameters of local models to update the global model. This process is effective and easy to implement, which establishes the fundamental framework for FL.

Since data is acquired from different devices in diverse environments, each local dataset inherently exhibits variations in the underlying distribution, *i.e.*, non-IID[1] (shifts due to weather conditions, night-time images, and various image degradations). Label distribution skew and feature distribution skew (see Definition 1 in §3.1) are two typical cases of non-IID situations in real-world

---

[*]Equal contribution.

[†]Lei Zhu (`leizhu@ust.hk`) is the corresponding author.

[1]We use "heterogeneous" and "non-IID" interchangeably.

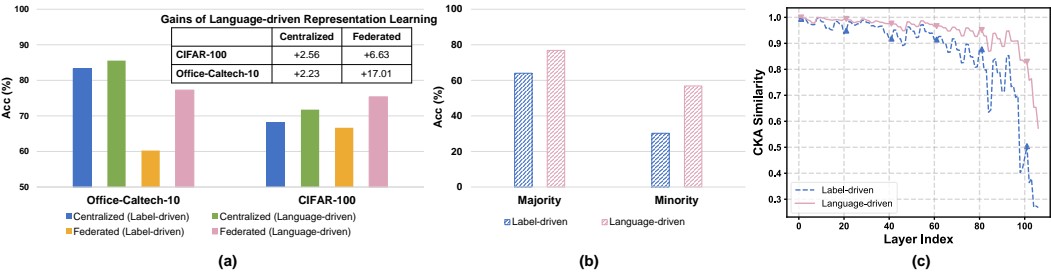

Figure 1: **Analysis of two types of training paradigm, *i.e.*, label-driven representation learning and language-driven representation learning**. **(a)** Accuracy comparisons on centralized and federated scenarios, where the language-driven representation learning improves Accs on the federated scenario higher than it improves on centralized learning. **(b)** Accuracy comparisons for both local majority and minority classes on CIFAR-100. Language-driven representation learning showed significant improvement over label-driven representation learning, especially for minority classes, indicating that language-driven representation learning can effectively enhance the generalization performance of local models. **(c)** Variations in CKA similarity (Kornblith et al., 2019) reveal that language-driven representation learning reduces the disparity in feature representations among local models.

applications, manifesting as data imbalances at the label level and feature level (Li et al., 2022c; Zhang et al., 2022). Such statistical heterogeneity incurs a significant decline in the performance of FedAvg (Zhang et al., 2022; Luo et al., 2021; Karimireddy et al., 2020; Li et al., 2020b; Zhao et al., 2018), emerging as a fundamental challenge in FL. This has attracted significant attention, and many studies are dedicated to improving the *local training* (Li et al., 2021; Tan et al., 2022a; Luo et al., 2021; Zhang et al., 2022; Gao et al., 2022) or *model aggregation* (Li et al., 2020c; Wang et al., 2020; Li et al., 2023; Fallah et al., 2020) of FedAvg on non-IID data. Despite the progress, most existing FL methods still follow label-driven representation learning during local training, such as cross-entropy loss in classification tasks.

In this work, we argue that it is not advisable to adopt the label-driven training paradigm, as it leads to significant drift in local models. Drawing on visual recognition as the running example, as mentioned in previous studies, induces model bias towards imbalanced local class distribution in terms of label distribution skew setting (Zhang et al., 2022; Diao et al., 2024). The local model seeks to fit the majority classes (see Definition 2 in §3.1) of the private dataset as much as possible (see Figure 1 (b)), incurring differences in both the feature representation layers (backbone) and the classifier. It exhibits greater bias in the classifier compared to other layers (Luo et al., 2021). In terms of feature distribution skew setting, using cross-entropy as the local objective will result in the feature shift problem (Li et al., 2020c; 2023; Zhou & Konukoglu, 2023), thereby degrading the performance of the global model (see Figure 1 (c)). The shortcomings of label-driven representation learning prompt us to explore a new training paradigm to redefine local training.

With the emergence of CLIP (Radford et al., 2021), language-driven representation learning has gained considerable attention and achieved success in various applications (Cha et al., 2023; He et al., 2023; Wang et al., 2022). Models trained with language showcase better generalization. Since language-driven representation learning is so effective in the centralized setting, we are thus interested in *whether language-driven representation learning can be a better training paradigm for FL, particularly in alleviating the non-IID issue.* By introducing it into FL, we surprisingly observed consistent accuracy improvements in FedAvg across two federated scenarios as shown in Figure 1 (a). We also observed that these improvements are significantly more pronounced in FL compared to centralized learning. This suggests that language-driven representation learning provides additional benefits to FedAvg in the federated setting. With deeper analysis, we further found that: Language-driven representation learning benefits **1**) *learning a unified feature representation across different clients* (see Figure 1 (c)); **2**) *alleviating the overfitting to majority classes* (see Figure 1 (b)), more details can be seen in *Appendix* §A. This explains why language-driven representation learning is more effective compared to traditional label-driven representation learning in heterogeneous FL scenarios.

Based on the above analyses, we propose **Fed**erated **G**lobal **L**anguage-image **C**ontrastive **L**earning, termed as `FedGLCL`. FedGLCL focused on redefining the local training phase of FedAvg, yielding a

language-driven training manner for FL on non-IID data. It instantiates the idea of language-driven representation learning by aligning each local image feature space to a common global text feature space during the training phase. Different from CLIP, we use a fixed pre-trained text encoder to encode the global class texts, thereby yielding a consistent text feature space. Subsequently, we use contrastive learning to train the image encoder from scratch. In our framework, there is no traditional classifier. Instead, similar to CLIP, we use similarity measures between images and text for predictions. Accordingly, we argue that the bias across different local models in our method, no matter whether caused by the label distribution skew or feature distribution skew, can be interpreted as the difference in feature representation. This issue has been alleviated through the alignment of feature spaces. Therefore, FedGLCL can be applied to various non-IID scenarios. Extensive experiments on four benchmarks demonstrated the effectiveness of our approach, which shows significant improvements under different non-IID scenarios.

**Contribution.** In this paper, we systematically explore the importance of language-driven representation learning for heterogeneous FL, yielding a language-driven FL framework, *i.e.*, FedGLCL. While seems straightforward, we believe this study provides more insights into addressing the non-IID issue in FL. Moreover, we offer comprehensive theoretical and empirical analyses to gain a deeper understanding of our approach, providing valuable insights for future research work.

## 2 RELATED WORK

### 2.1 FEDERATED LEARNING ON NON-IID DATA

How to alleviate the impact of non-IID data on model training has always been a critical issue in the field of FL (Ye et al., 2023; Huang et al., 2023; Luo et al., 2021; Feng et al., 2023b). Variations in the distribution of data across different clients may result in optimization bias in the local models, thereby significantly degrading the performance of the model aggregation (Karimireddy et al., 2020; Zhao et al., 2018). To address this, a type of solution (Durmus et al., 2021; Li et al., 2020b; Zhang et al., 2022; Li et al., 2021; Gao et al., 2022) is to minimize the optimization difference among local models during the local training phase. For example, FedProx (Li et al., 2020b) and FedDyn (Durmus et al., 2021) introduced additional regulation terms into local objectives to mitigate the bias between local models. In contrast to these approaches, another type of solution. (Li et al., 2020c; Hsu et al., 2019; Wang et al., 2020; Fallah et al., 2020; Ma et al., 2022; Lee et al., 2023) focuses on designing novel model aggregation strategies to yield a better global model. For instance, Fed-Nova (Wang et al., 2020) proposed a normalized averaging method, which corrects local model parameters before averaging. Besides, a typical representative among these methods is personalized aggregation strategies (Li et al., 2020c; Arivazhagan et al., 2019; Collins et al., 2021; Lee et al., 2023), which mitigate the impact of data heterogeneity by allowing local models to retain a portion of personalized parameters.

In this work, we focus on the local training phase. While previous studies have made progress on this type of solution, most of them still use label-driven representation learning as the local training paradigm. They typically employ cross-entropy as the primary supervisory training loss and additionally incorporate some auxiliary losses to mitigate bias in local models, such as regulation (Li et al., 2020b; Durmus et al., 2021), distillation (Yao et al., 2023; Lee et al., 2022) and prototype learning (Tan et al., 2022a;b). While effective, it is unable to overcome the inherent disadvantages of label-driven training (see Figure 1 (b) and (c)). Given this, this work redefines the process of local training by language-driven representation learning, which can effectively address non-IID issues.

### 2.2 LANGUAGE-DRIVEN REPRESENTATION LEARNING

Recently, language-driven representation learning (Gan et al., 2022) has been a very hot topic in the field of vision, giving rise to numerous vision-language models, such as CLIP (Radford et al., 2021) and Align (Jia et al., 2021). CLIP employs text and image encoders to learn rich semantic representations from extensive pairs of images and texts by a contrastive loss. This grants CLIP more powerful feature representation capabilities and superior zero-shot learning performance, prompting lots of studies to directly apply it to various downstream tasks (Zhou et al., 2022; Patashnik et al., 2021; Wysoczańska et al., 2024; Luo et al., 2022; Gao et al., 2021). Besides direct application, another is to modify the text-image feature alignment approach of CLIP for specific downstream

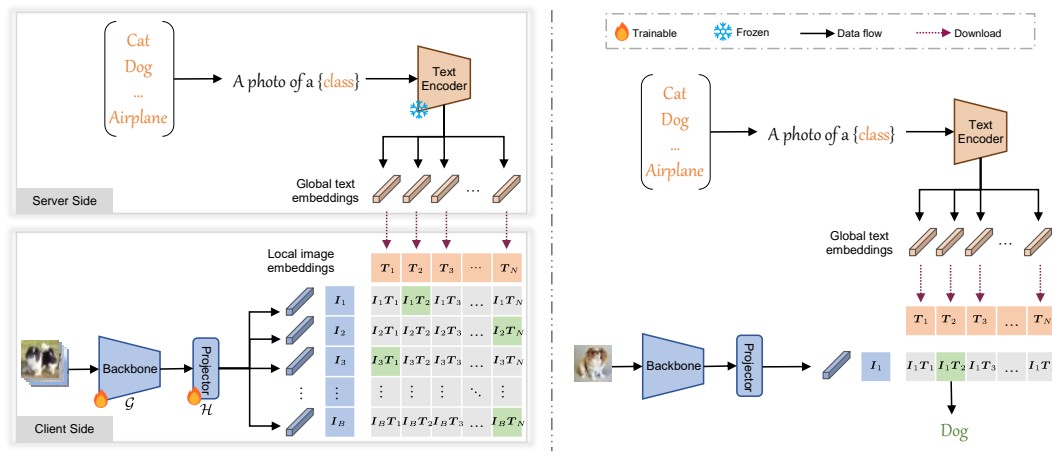

(a) Training Stage        (b) Inference Stage

Figure 2: **Illustration of FedGLCL.** (a) During the training stage, FedGLCL utilizes a frozen text encoder to embed the global class texts on the server side, thereby yielding a unique global text feature space. Then, each client aligns its own local image feature space with the global text feature space. We omit the model aggregation procedure, as it is identical to FedAvg. (b) During the inference stage, we make predictions based on the similarity between images and text.

tasks (Li et al., 2022a; Xu et al., 2022; Wang et al., 2023). For example, RegionCLIP (Li et al., 2022b) and GLIP (Cha et al., 2023) proposed region-text alignment to learn region-level semantic information for object detection. The idea of region-text alignment has also been applied in semantic segmentation tasks (Cha et al., 2023; Liang et al., 2023).

However, little is understood about the importance and applicability of language-driven representation learning for FL. Previous studies have merely applied the pre-trained CLIP model directly to FL as a foundational component (Lu et al., 2023; Guo et al., 2023; Shi et al., 2023), which is unrelated to language-driven representation learning. In this paper, we thoroughly explored language-driven representation learning for non-IID FL settings and proposed a novel language-driven FL framework.

## 3 METHODOLOGY

### 3.1 PRELIMINARY

**Federated Learning.** Suppose there are $K$ clients and each client $k \in [K]$ has a set of supervised training instances $\{(x_i, y_i)\}_{i=1}^{n_k}$. The image $x_i \in \boldsymbol{X}$ and label $y_i \in \boldsymbol{Y}$ are from a device-indexed joint distribution $(x_i, y_i) \sim \mathbb{P}_k(x, y)$. The goal of FL is to collaboratively learn a model $f$ without sharing local private data, which can be formulated as the following optimization problem:

$$\mathcal{L}(\boldsymbol{w}) = \sum_{k=1}^{K} \gamma_k \mathcal{L}_k(\boldsymbol{w}), \quad \text{where} \quad \gamma_k = \frac{n_k}{\sum_{i=1}^{K} n_i}. \tag{1}$$

To solve this, FedAvg (McMahan et al., 2017) minimizes each local empirical loss $\mathcal{L}_k(\boldsymbol{w})$ to parallel train local models — *local training* at the clients and then average the updated local model parameters — *model aggregation* at the server:

$$\mathcal{L}_k(\boldsymbol{w}_k^t) = \frac{1}{n_k} \sum_{(x_i, y_i) \sim \mathbb{P}_k} \ell(f(x_i; \boldsymbol{w}_k^{t-1}); y_i), \tag{2}$$

$$\boldsymbol{w}_G^t = \sum_{k=1}^{K} \gamma_k \boldsymbol{w}_k^t, \tag{3}$$

where the superscript $t$ represents the model parameters after $t$-th round training, $\boldsymbol{w}_k$ and $\boldsymbol{w}_G$ are the parameters of local model and global model. In label-driven FL, the loss function $\ell$ is usually a cross-entropy loss for the image classification task.

---

**Algorithm 1:** FedGLCL

---

**Input:** Number of Clients $K$, communication rounds $M$, epochs $E$, learning rate $\eta$, temperature $\tau$
**Output:** $\boldsymbol{w}_G^M$

1  **Server-side Execution:**
2  $\boldsymbol{T} \leftarrow$ Embed global class texts
3  **for** *round* $t = 1, 2, ..., M$ **do**
4     **for** *client* $k = 1, 2, ..., K$ **parallelly do**
5        **for** $(x_i, y_i) \sim \mathbb{P}_k$ **do**
6           $\boldsymbol{w}_k^t \leftarrow$ **Local Training** $(k, \boldsymbol{w}_G^{t-1}, \boldsymbol{T})$
7        **end**
8     **end**
9     $\boldsymbol{w}_G^t \leftarrow \sum_{k=1}^K \gamma_k \boldsymbol{w}_k^t$
10 **end**
11 **Local Training** $(k, \boldsymbol{w}_G^{t-1}, \boldsymbol{T})$:
12 $\boldsymbol{w}_k^t \leftarrow \boldsymbol{w}_G^{t-1}$
13 **for** *epoch* $e = 1, 2, ..., E$ **do**
14    **for** $(x_i, y_i) \sim \mathbb{P}_k$ **do**
15       $\boldsymbol{I}_i \leftarrow f(x_i; \boldsymbol{w}_k^t)$
16       $\delta_i \leftarrow \frac{\boldsymbol{I}_i \cdot \boldsymbol{T}^\top}{\|\boldsymbol{I}_i\| \|\boldsymbol{T}\|}$
17       $\ell \leftarrow -\log \frac{\exp(\delta_{i,y_i}/\tau)}{\sum_{j \neq y_i} \exp(\delta_{i,j}/\tau)}$
18       $\boldsymbol{w}_k^t \leftarrow \boldsymbol{w}_k^t - \eta \nabla \ell$
19    **end**
20 **end**
21 return $\boldsymbol{w}_k^t$ to server

---

**Definition 1 (Non-IID Data In Federated Learning)** *The local distribution $\mathbb{P}_k(x, y)$ can be rewritten as $\mathbb{P}_k(y|x)\mathbb{P}_k(x)$ or $\mathbb{P}_k(x|y)\mathbb{P}_k(y)$. For label distribution skew, $\mathbb{P}_k(y)$ varies across clients while $\mathbb{P}_k(x|y)$ is consistent for all clients. In contrast, for feature distribution skew, $\mathbb{P}_k(y|x)$ keeps consistent for all clients but $\mathbb{P}_k(x)$ is different.*

**Definition 2 (Majority and Minority Classes)** *For label distribution skew, the number of instances for each class on the client side is imbalanced. We partition the $\boldsymbol{Y}$ into majority classes $\boldsymbol{Y}_\phi$ and minority classes $\boldsymbol{Y}_\psi$ based on whether their quantities exceed a threshold $\tau$, where $\boldsymbol{Y} = \boldsymbol{Y}_\phi \cup \boldsymbol{Y}_\psi$.*

Tradition FL methods adopt label-driven representation learning as the local training paradigm, which struggles to exhibit robustness on non-IID data, leading to inconsistent local and global objectives, *i.e.*, the global model from Eq. (3) deviate from the solution of Eq. (1), termed *client drift*. Previous studies (Li et al., 2020b; Durmus et al., 2021; Li et al., 2021; Lee et al., 2022) attempt to introduce various forms of auxiliary loss terms into Eq. (2), but it remains challenging to fully overcome the inherent limitations of label-driven representation learning for client drift (Karimireddy et al., 2020). In this work, we explore a novel federated global language-image contrastive learning, namely FedGLCL, to address the non-IID issue in FL.

## 3.2 FEDERATED GLOBAL LANGUAGE-IMAGE CONTRASTIVE LEARNING

Our approach, `FedGLCL`, is the first language-driven federated learning framework, which alleviates the non-IID from a feature perspective. Language-driven training paradigm typically involves a text encoder and an image encoder, mapping text and images to their respective feature spaces. In FedGLCL, we have deployed a text encoder on the server side, while each client has an image encoder and a projector. The key is to embed local images and global texts into a common space, learning a consistent feature presentation among clients. Meanwhile, the learned metric between image and text features enables making predictions. We illustrate the training and inference of our framework in Figure 2 and describe each part in detail below.

**Global Text Embeddings.** If training text encoders independently on each local client, the imbalanced data distribution will lead to multiple skewed local text feature spaces, which runs counter to our objectives (see §5.5). To achieve consistent global text feature spaces, we use a pre-trained text encoder to embed the set of $N$ texts, producing $N$ text embeddings $\boldsymbol{T} = \{\boldsymbol{T}_1, \boldsymbol{T}_2, \ldots, \boldsymbol{T}_N\} \in$

$\mathbb{R}^{N \times V}$. The above process is carried out solely on the server side before training, and previous work (Li et al., 2021; 2020c; Zhou & Konukoglu, 2023; Diao et al., 2024; Zhang et al., 2022) typically assumes that the complete set $\boldsymbol{Y}$ of classes is known. The class is embedded into a context prompt such as "a photo of a {class}", where "class" is the class name. The global text embeddings will be transferred to all clients for local training.

**Local Training.** On the $k$-th client, the local model $f$ has an image encoder $\mathcal{G}$ serves as the backbone to extract the image features, and a lightweight projector $\mathcal{H}$ to align the dimensions of image and text features:

$$f = \mathcal{H} \circ \mathcal{G}, \quad \boldsymbol{w}_k = \boldsymbol{w}_k^{\mathcal{H}} \cup \boldsymbol{w}_k^{\mathcal{G}}, \tag{4}$$

where $\boldsymbol{w}_k^{\mathcal{G}}$ and $\boldsymbol{w}_k^{\mathcal{H}}$ are the parameters of backbone and projector. Given a training instance $(x_i, y_i) \sim \mathbb{P}_k$, where $x_i \in \mathbb{R}^{C \times H \times W}$ is an image with spatial size $(H \times W)$ and $C$ channels, $y_i \in [N]$ is the label, we can get the corresponding image embedding as follows:

$$\boldsymbol{I}_i = f(x_i; \boldsymbol{w_k}) \quad \in \mathbb{R}^V. \tag{5}$$

Following this, we can get the similarity between image embedding and global text embeddings as follows:

$$\delta_i = \frac{\boldsymbol{I}_i \cdot \boldsymbol{T}^\top}{\|\boldsymbol{I}_i\|\|\boldsymbol{T}\|} \quad \in \mathbb{R}^N. \tag{6}$$

To align the image and text into a common space, we introduce contrastive learning (Hadsell et al., 2006; Oord et al., 2018). Consider the image embedding $\boldsymbol{I}_i$ as a query, its positive key is the text embedding $T_{y_i}$ corresponding to class $y_i$. The rest of the $N - 1$ text embeddings are considered negative keys for query, which should be dissimilar to $\boldsymbol{I}_i$. Thus, the loss function $\ell$ of Eq. (2) can be rewritten as:

$$\ell = -\log \frac{\exp(\delta_{i,y_i}/\tau)}{\sum_{j \in [N], j \neq y_i} \exp(\delta_{i,j}/\tau)}, \tag{7}$$

where $\tau$ is the temperature parameter that can control the tolerance for feature differences (Wu et al., 2018; Zhang et al., 2021). Notably, only the backbone and projector are trainable in our framework. At the end of local training in each round, clients upload their updated parameters of backbone and projector to the server, and we can update the global model by Eq. (3).

Algorithm 1 shows the detailed training procedure of FedGLCL. Before training, the server first encodes the class names using a pre-trained encoder to acquire global text embeddings, which are subsequently transferred to individual local clients. During the local training stage, each client learns the consistent image feature representations with the guide of global text embeddings. By this, the differences among local models will be decreased, leading to a better global model. Our model aggregation process aligns with FedAvg (McMahan et al., 2017).

**Language-driven Inference.** After optimization through contrastive learning, the trained backbone and projector exhibit strong feature representation capabilities. Intuitively, their image embeddings are closer to the text embeddings of the corresponding class as far as others. Meanwhile, these text embeddings contain class-wise information, which can be used to distinguish different classes. The learned metrics between image and text can be used for making predictions. Therefore, similar to CLIP, we can use the similarity between image and text embeddings for prediction. The process of prediction for a given image $\boldsymbol{x}$ can be described as follows:

$$\widetilde{y} = \arg\max \ \text{softmax}(\frac{\boldsymbol{I} \cdot \boldsymbol{T}^\top}{\|\boldsymbol{I}\|\|\boldsymbol{T}\|}), \quad \text{and} \quad \boldsymbol{I} = \mathcal{H}(\mathcal{G}(x; \boldsymbol{w}_G^{\mathcal{G}}); \boldsymbol{w}_G^{\mathcal{H}}), \tag{8}$$

where $\boldsymbol{w}_G^{\mathcal{G}}$ and $\boldsymbol{w}_G^{\mathcal{H}}$ are the optimized parameters of global backbone and projector.

## 4 THEORETICAL ANALYSIS

In this section, we provide deep insights into our framework by theoretical analysis. To understand the underlying working principle of our method, we provide the following three theorems.

**Theorem 1.** *Consider a FL system with $K$ clients, and $K$ training instances $\{(x_k, \sigma)_{k=1}^K\}$ from each client, satisfy $(x_k, \sigma) \sim \mathbb{P}_k$. Let $\boldsymbol{T} = \{\boldsymbol{T}_1, \dots, \boldsymbol{T_N}\}$ be a set of $N$ text embeddings containing one positive sample and $N - 1$ negative samples, local image embedding $I_k = f(x_k; \boldsymbol{w}_k)$. Then,*

*the total mutual information between $K$ local image embeddings and their positive sample $\boldsymbol{T}_\sigma$ has a lower bound*

$$\sum_{k=1}^{K} \mathcal{M}(\boldsymbol{I}_k, \boldsymbol{T}_\sigma) \geq K \log(N) - \sum_{k=1}^{K} \mathcal{L}_k. \tag{9}$$

**Theorem 2.** *In FedGLCL, the loss function $\mathcal{L}_k$ of client $k$ can be expressed as:*

$$\mathcal{L}_k = \ell^+(x_i; \boldsymbol{T}_{y_i}; \boldsymbol{w}_k) + \ell^-(x_i; \{\boldsymbol{T}_j\}_{j \in [N], j \neq y_i}; \boldsymbol{w}_k), \tag{10}$$

*where $\ell^+$ is the target objective, and $\ell^-$ is the regulation term to prevent overfitting.*

Theorem 1 indicates that FedGLCL aims to increase the mutual information between local feature representations and global text embeddings. This ensures that all local feature representations converge towards a common objective, reducing the disparity in feature representations. Besides, Theorem 2 further shows that FedGLCL introduces regulation effects into local training, thereby preventing overfitting to the majority classes.

**Theorem 3.** *Let $\mathbb{P}_1, \mathbb{P}_2, \ldots, \mathbb{P}_K$ be the empirical data distribution and $\hat{\mathbb{P}}_1, \hat{\mathbb{P}}_2, \ldots, \hat{\mathbb{P}}_K$ be the true data distribution. Denote the projector $\mathcal{H}$ as the hypothesis from hypothesis space $\hat{\mathcal{H}}$ and $d$ be the VC-dimension of $\hat{\mathcal{H}}$. The number of text embeddings is $N$. With probability at least $1 - \delta$,*

$$\max_{(\boldsymbol{w}_1, \boldsymbol{w}_2, \ldots, \boldsymbol{w}_K)} \left| \sum_{k=1}^{K} \gamma_k \mathcal{L}_{\mathbb{P}_k}(\boldsymbol{T}; \boldsymbol{w}_k) - \sum_{k=1}^{K} \gamma_k \mathcal{L}_{\hat{\mathbb{P}}_k}(\boldsymbol{T}; \boldsymbol{w}_k) \right|$$

$$\leq \sqrt{\frac{\sum_{k=1}^{K} n_k}{2} \log \frac{N}{\delta}} + \sqrt{\frac{d}{\sum_{k=1}^{K} n_k} \log \frac{e \sum_{k=1}^{K} n_k}{d}}. \tag{11}$$

Theorem 3 provides the bound of our generalization error and indicates our method can achieve the expected performance. The detailed analysis and proof are provided in the *Appendix* §B.

## 5 EXPERIMENTS

### 5.1 EXPERIMENTAL SETUP

**Datasets.** We conduct experiments on two different non-IID FL scenarios, including `label distribution skew`: **CIFAR-100** (Krizhevsky et al., 2009) and **Fashion-MNIST** (Xiao et al., 2017), and `feature distribution skew`: **Office-Caltech-10** (Gong et al., 2012) and **DomainNet** (Peng et al., 2019). For label distribution skew, following (Li et al., 2021; Tan et al., 2022a; Diao et al., 2024), we randomly partition datasets according to the Dirichlet distribution ($\mathbb{P}_k \sim Dir(\beta)$) or select $\mathbb{C}$ classes for each client. The number of clients is 10 and all participate in training, *i.e.*, the sample fraction is 1. For feature distribution skew, we use the subsets from different domains as clients (Zhou & Konukoglu, 2023; Li et al., 2020c), where the number of clients in Office-Caltech-10 and DomainNet are 4 and 6, respectively.

**Baselines.** We compare FedGLCL with several state-of-the-art FL methods for non-IID data, including **FedAvg** (McMahan et al., 2017), **FedProx** (Li et al., 2020b), **FedNova** (Wang et al., 2020), **Scaffold** (Karimireddy et al., 2020), **MOON** (Li et al., 2021), and **FedProto** (Tan et al., 2022a), which are popular baselines in heterogeneous FL. Moreover, we compared some FL methods that are designed for specialized scenarios, such as **FedBN** (Li et al., 2020c) and **FedFA** (Zhou & Konukoglu, 2023) for the feature distribution skew scenario, and **FedLC** (Zhang et al., 2022) and **FedConcat** (Diao et al., 2024) for the label distribution skew scenario.

**Implementation Details.** We implement our method and other baselines by Pytorch and conduct all experiments on an NVIDIA RTX 4090 GPU with 24 GB memory. We adopt AlexNet (Krizhevsky et al., 2012) for Office-Caltech-10, DomainNet and Fashion-MNIST, and ResNet-50 (He et al., 2016) for CIFAR-100. The models are trained with the SGD optimizer, with a learning rate of 0.0001 for Office-Caltech-10 and DomainNet, and 0.1 for Fashion-MNIST and CIFAR-100. Additionally, the batch size is set to 64 for CIFAR-100, 32 for Office-Caltech-10 and DomainNet, and 256 for Fashion-MNIST. The local epoch is set to 5. For a fair comparison, all methods use the above

Table 1: **Test accuracy (%) on feature distribution skew scenario.** For a detailed comparison, we present the test accuracy of each client, *i.e.*, **Office-Caltech-10** (Gong et al., 2012): A(Amazon), C(Caltech), D(DSLR), W(Webcam), **DomainNet** (Peng et al., 2019): C(Clipart), I(Infograph), P(Painting), Q(Quickdraw), R(Real), S(Sketch). **Avg.** denotes the average accuracy of all clients.

| Method | Office-Caltech-10 | | | | | DomainNet | | | | | | |
|---|---|---|---|---|---|---|---|---|---|---|---|---|
| | A | C | D | W | Avg. | C | I | P | Q | R | S | Avg. |
| FedAvg | 50.00 | 49.77 | 53.12 | 88.13 | 60.25 | 75.47 | 37.44 | 64.45 | 73.10 | 73.04 | 74.18 | 66.28 |
| FedProx | 57.81 | 46.66 | 56.25 | 76.27 | 59.25 | 75.28 | 36.68 | 62.68 | 71.20 | 68.94 | 71.84 | 64.43 |
| FedNova | 50.00 | 42.22 | 62.50 | 88.13 | 60.71 | 75.85 | 35.00 | 64.94 | 70.60 | 70.66 | 72.56 | 64.93 |
| Scaffold | 45.83 | 39.55 | 67.75 | 76.66 | 58.45 | 69.20 | 36.37 | 57.67 | 55.20 | 66.88 | 60.10 | 57.57 |
| MOON | 53.64 | 44.88 | 53.12 | 89.83 | 60.37 | 72.24 | 35.15 | 65.91 | 58.10 | 72.47 | 64.07 | 61.32 |
| FedProto | 52.08 | 40.88 | 78.12 | 86.44 | 65.23 | 76.42 | 22.83 | 60.90 | 89.60 | 79.86 | 70.03 | 66.61 |
| FedBN | 60.93 | 44.44 | 84.37 | 86.44 | 69.04 | 81.36 | 29.68 | 63.97 | 89.60 | 82.82 | 73.10 | 70.09 |
| FedFA | 61.45 | 49.77 | **84.37** | 89.83 | 71.36 | 83.27 | 20.24 | 71.40 | **91.40** | **87.75** | **83.57** | 72.94 |
| **FedGLCL (Ours)** | **75.52** | **64.00** | 75.00 | **94.91** | **77.35** | **84.60** | 37.44 | **79.96** | 88.50 | 86.27 | 82.49 | **76.54** |

Table 2: **Test accuracy (%) on label distribution skew scenario.** We evaluate all methods on the `test` of CIFAR-100 (Krizhevsky et al., 2009) and Fashion-MNIST (Xiao et al., 2017) with different partition strategies, where smaller $\beta$ and $\mathbb{C}$ means higher heterogeneity.

| Method | CIFAR-100 | | | Fashion-MNIST | | | | |
|---|---|---|---|---|---|---|---|---|
| | $\beta = 0.1$ | $\beta = 0.5$ | $\beta = 1.0$ | $\mathbb{C} = 2$ | $\mathbb{C} = 3$ | $\beta = 0.3$ | $\beta = 0.5$ | $\beta = 1.0$ |
| FedAvg | 64.52 | 66.67 | 67.75 | 42.69 | 74.50 | 89.46 | 92.35 | 92.65 |
| FedProx | 63.98 | 68.64 | 69.21 | 38.98 | 74.04 | 87.84 | 92.11 | 92.51 |
| FedNova | 63.19 | 68.34 | 68.78 | 50.07 | 60.87 | 90.13 | 92.49 | 92.80 |
| Scaffold | 59.61 | 68.77 | 70.89 | 29.44 | 57.84 | 91.45 | 92.44 | 92.95 |
| MOON | 66.30 | 70.91 | 71.85 | 51.95 | 74.40 | 88.96 | 91.00 | 91.36 |
| FedProto | 66.64 | 70.04 | 70.51 | 43.95 | 72.82 | 88.64 | 91.92 | 92.33 |
| FedLC | 63.34 | 65.70 | 66.91 | 37.28 | 71.06 | 86.49 | 91.66 | 92.08 |
| FedConcat | 67.50 | 71.82 | 72.49 | 52.29 | 75.65 | 89.83 | 92.20 | 92.47 |
| **FedGLCL (Ours)** | 68.58 | 73.52 | 74.85 | 56.63 | 84.47 | 93.04 | 93.82 | **93.98** |
| **FedGLCL (Ours) + FedProx** | 68.95 | 74.02 | 75.15 | 58.69 | **85.28** | 93.34 | **93.86** | 93.93 |
| **FedGLCL (Ours) + MOON** | **69.36** | **74.38** | **75.56** | **59.71** | 84.61 | **93.36** | 93.85 | 93.89 |

settings. We use the popular language model, *i.e.*, Bert-base (Devlin et al., 2018) as text encoder and $V$ is 640. The temperature is 0.07 as default (Wang & Liu, 2021). Notably, the image backbone is trained from scratch without loading any pre-trained models. More details and experimental results are presented in the *Appendix* §C and §D.

## 5.2 MAIN RESULTS

In this section, we present the overall results of all methods on four popular FL benchmarks under two different scenarios: feature distribution skew in Table 1 and label distribution skew in Table 2. The performance of the method is quantified using Top-1 accuracy.

**Comparison with State-of-the-arts.** As we can see, FedGLCL consistently outperforms all baselines across different scenarios, achieving new state-of-the-art performance. In particular, by making lightweight modifications to the local training of FedAvg, our FedGLCL yields significant improvements over FedAvg across four different datasets, *e.g.*, Office-Caltech-10: 60.25% → **77.35** %, DomainNet: 66.28% → **76.54**%, CIFAR-100: 66.67% → **73.52**%, and Fashion-MNIST: 42.69% → **56.63**%. This indicates the effectiveness of language-driven representation learning in mitigating data heterogeneity in federated learning. Besides, the methods designed for specific scenarios demonstrate their advantages in the corresponding setting, achieving better performance than other general heterogeneous FL frameworks. For example, FedFA outperforms other baselines in the feature distribution skew scenario while FedGLCL yields higher accuracy than other baselines in the label distribution skew. However, both of them achieve worse performance them our FedGLCL,

which shows the robustness and generality of our method for different non-IID scenarios. Moreover, our FedGLCL also significantly outperforms the other baselines focused on local training such as FedProx, MOON and FedProto. This indicates that global text embeddings possess a stronger guiding capability than model relations (FedProx and MOON) or class prototypes (FedProto). It is worth noting that FedLC, which proposed a calibrated cross-entropy loss, achieves worse performance than FedAvg, consistent with the results in the previous study (Diao et al., 2024). This highlights the inherent limitation of label-driven representation learning in non-IID scenarios. In contrast, language-driven representation learning exhibits more generalized and robust performance.

**Combined with Other Baselines.** The core of FedGLCL lies in using language-driven representation learning to replace the traditional label-driven training paradigm. This means that FedGLCL can substitute for the supervised loss component in the local objective of previous FL methods, *i.e.*, cross-entropy loss. To explore this, we combine our approach with FedProx and MOON and the loss weight is unchanged. Their accuracy is presented in Table 2. Notably, after incorporating our language-representation learning as the local training manner, the performance of FedProx and MOON has significantly improved compared to their use of traditional label supervision. Surprisingly, we find the approximation term in FedProx and the model contrastive loss term in MOON also contribute to the performance improvement of FedGLCL, which provides additional guidance information for language-driven training.

## 5.3 SCALABILITY

To investigate the scalability of FedGLCL, we build up different scales of federation by splitting the dataset into more clients on CIFAR-100 (Li et al., 2022c; Diao et al., 2024). There are two different settings: (1) **Full**: we increase the number of clients from 10 to 30 while keeping the sample fraction at 1. (2) **Partial**: we set the number of clients and sample fraction to (20, 0.5) and (100, 0.1), respectively, keeping the number of clients participating in each round of training the same as the default setting (10 training clients per round). Other experimental settings remain the same as the default configuration. As shown in Table 3, all methods exhibit a decline in performance as the number of clients in-

Table 3: **Test accuracy (%) with different number of clients (#K, #SF)** on CIFAR-100 under two different participation setting, *i.e.*, **Full** and **Partial**, where **#K** is the total number of clients, and **#SF** is the sample fraction of training clients per round. The $\beta$ is 0.5.

| Method | Full | | | Partial | |
|---|---|---|---|---|---|
| | (10, 1) | (20, 1) | (30, 1) | (20, 0.5) | (100, 0.1) |
| FedAvg | 66.67 | 65.32 | 64.24 | 64.39 | 39.22 |
| FedProx | 68.04 | 66.02 | 65.30 | 65.32 | 40.78 |
| FedNova | 68.34 | 67.36 | 63.68 | 63.97 | 37.81 |
| Scaffold | 68.77 | 66.12 | 65.38 | 64.22 | 39.75 |
| MOON | 70.91 | 67.30 | 65.85 | 65.36 | 42.54 |
| FedProto | 70.04 | 69.44 | 66.61 | 63.24 | 38.32 |
| FedLC | 65.70 | 64.20 | 62.95 | 60.49 | 37.65 |
| FedConcat | 71.82 | 70.02 | 68.77 | 66.05 | 44.98 |
| **FedGLCL (Ours)** | **73.52** | **71.71** | **70.20** | **69.66** | **49.17** |

creases, but FedGLCL still outperforms other baselines. This suggests that our language-driven training paradigm is resilient to changes in the number of clients and can be seamlessly deployed in larger federations, even under partial participation settings.

## 5.4 EFFICIENCY ANALYSIS

**Communication Cost.** Since the size of global text embeddings is $N \times 640$, which is smaller the dimension of the classifier layer, *i.e.*, $N \times 2048$ for ResNet-50 and $N \times 4096$ for AlexNet. Therefore, FedGLCL does not incur additional communication overhead, and its communication overhead is even slightly lower than that of FedAvg. For example, FedGLCL reduces the communication overhead per round by **0.12M** on CIFAR-100 compared to FedAvg. This highlights the advantage of FedGLCL over traditional label-driven representation learning in terms of communication efficiency.

**Communication Efficiency.** In Figure 3, we show the test accuracy of several FL frameworks versus communication rounds on two scenarios. Our FedGLCL exhibits a faster convergence speed than FedAvg in two data heterogeneity scenarios, which further indicates the superiority of language-

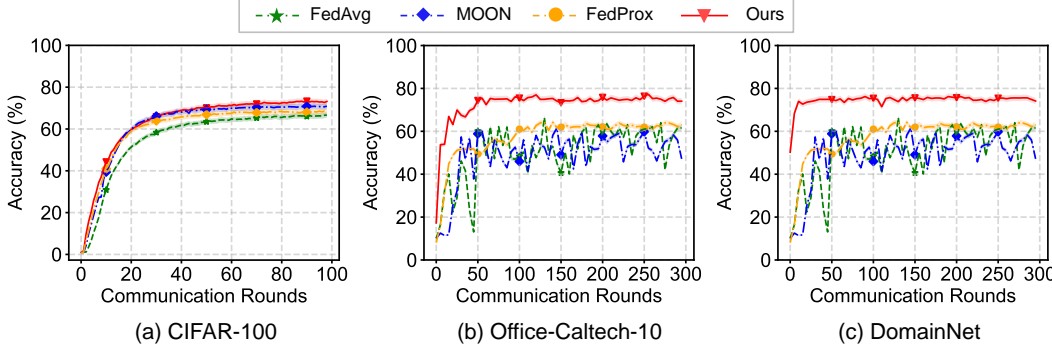

Figure 3: **Test accuracy (%)** of several FL frameworks versus communication rounds on three benchmarks. The $\beta$ is 0.5 for CIFAR-100 (Krizhevsky et al., 2009).

driven representation learning. Compared with FedProx and MOON, which also emphasize improving local training, FedGLCL achieves a faster and more stable convergence on Office-Caltech-10 and DomainNet. It is worth noting that FedGLCL achieves high accuracy with just one communication round on DomainNet, a larger dataset for feature distribution skew. This suggests that the alignment of features between local images and global text in FedGLCL can effectively mitigate the feature shift issue.

## 5.5 ABLATION STUDY

**FedGLCL vs. FedGLCL-L.** Global text embedding is a core insight of our method. To further validate our insights, we construct `FedGLCL-L`: each client has a trainable text encoder optimized by Eq. (7), and after each round of local training, their encoders are averaged to obtain a globally trainable text encoder. The results in Table 4 indicate that skewed local

Table 4: **Ablation studies** of FedGLCL on three benchmarks. The $\beta$ is 0.5 for CIFAR-100 (Krizhevsky et al., 2009).

| Variant | CIFAR-100 | Office-Caltech-10 | DomainNet |
|---|---|---|---|
| FedAvg | 66.67 | 60.25 | 66.28 |
| FedGLCL-L | 63.47 | 56.80 | 58.75 |
| **FedGLCL** | **73.52** | **77.35** | **76.54** |

text feature spaces fail to assist clients in learning consistent image feature representations, consequently resulting in significant performance degradation.

**Hyper-parameter Analysis.** FedGLCL involves only one hyper-parameter, *i.e.*, temperature $\tau$. To explore its fluence for our method, we tune $\tau$ from $\{0.01, 0.07, 0.1, 0.5, 1\}$ on CIFAR-100 with $\beta = 0.5$ and shows the results in Figure 4. We observe that a smaller value for $\tau$ leads to better performance, aligning with a common belief about $\tau$ in contrastive learning. This is because a smaller temperature coefficient places more emphasis on distinguishing challenging samples that are similar to the anchor, often resulting in more uniform representations (Wang & Liu, 2021).

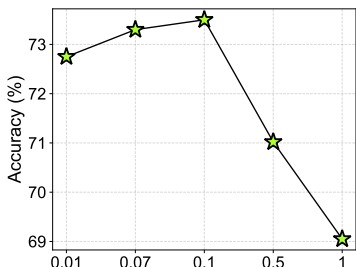

Figure 4: **Hyper-parameter analysis** for $\tau$ on CIFAR-100 (Krizhevsky et al., 2009) under $\beta = 0.5$.

## 6 CONCLUSION

In this work, we focused on addressing the non-IID issue, a fundamental challenge in FL. Due to inherent limitations in label-driven FL, this paper explores the applicability of language-driven representation learning in FL. We found that language-driven representation learning can benefit local models to learn consistent representations and prevent overfitting to majority classes, effectively addressing the non-IID problem, and yielding a novel FL framework, *i.e.*,FedGLCL. The advantage of language-driven representation learning allows it to exhibit outstanding performance across different scenarios. Finally, a comprehensive theoretical and empirical analysis helps us better understand FedGLCL, thereby providing more insights into the applications of language-driven representation learning in FL.

ACKNOWLEDGMENTS

This work is supported by the Guangzhou-HKUST(GZ) Joint Funding Program (No. 2023A03J0671), the National Research Foundation, Singapore under its AI Singapore Programme (AISG Award No: AISG2-TC-2021-003), Agency for Science, Technology and Research (A*STAR) through its AME Programmatic Funding Scheme Under Project A20H4b0141, A*STAR Central Research Fund "A Secure and Privacy Preserving AI Platform for Digital Health", and Agency for Science, Technology and Research (A*STAR) through its RIE2020 Health and Biomedical Sciences (HBMS) Industry Alignment Fund Pre-Positioning (IAF-PP) (grant no. H20C6a0032), and Guangdong Provincial Key Lab of Integrated Communication, Sensing and Computation for Ubiquitous Internet of Things(No.2023B1212010007).

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

## A    EMPIRICAL ANALYSIS

In this work, we found that language-driven representation learning has two inherent advantages for FL under non-IID data. First, *it effectively reduces the feature representation difference among local models.* We employ the Centered Kernel Alignment (Kornblith et al., 2019) (CKA) similarity to quantify the differences in feature representations among different local models. CKA is a reliable technique for quantifying the similarity between pairs of feature representations, widely used in various applications as an analytical tool (Raghu et al., 2021). Specifically, we use the global model as the anchor and calculate the CKA similarity for each layer of the network between all local models and the anchor. We then visualize the average CKA scores of all local models in Figure 1 (c). The results indicate that the representation disparity among local models under non-IID is primarily concentrated in the deep layers of the network (A larger index means a deeper layer), and language-driven representation learning can effectively mitigate these differences. This is facilitated by the text-image feature alignment of language-driven representation learning. Second, *it helps prevent local models from overfitting to the majority classes.* Label-driven representation learning can easily lead to overfitting on the majority classes due to a lack of supervised information on minority classes. In contrast, during the training process, language-driven representation learning not only brings the features of images closer to the corresponding textual features of their corresponding class but also pushes them away from the text features of other classes, thereby indirectly introducing supervisory information from other classes. This process effectively prevents the overfitting of local models and enhances their generalization performance (see Figure 1 (b)). Building upon the above two aspects, language-driven representation learning achieves consistent improvements in two different federated scenarios, and its enhancement for FedAvg is superior to that in centralized learning (see Figure 1 (a)). Motivated by this, we suggest using language-driven representation learning as the local training paradigm to address the non-IID issue.

## B    THEORETICAL ANALYSIS

### B.1    ANALYSIS OF WORKING PRINCIPLE

As stated in §1, language-driven representation learning exhibits two empirical phenomenons, *i.e.,* *mitigating feature difference* and *preventing overfitting*, which benefit for addressing the non-IID issue. In this section, we provide a theoretical analysis to understand the working principle of FedGLCL. We first explain why FedGLCL can reduce the feature representation differences among local models from the perspective of mutual information (Belghazi et al., 2018), which is a commonly used method to quantify and optimize the differences in feature representations. First, we introduce the following lemma from (Oord et al., 2018) for contrastive learning.

**Lemma 1.** *Given a set $\boldsymbol{X} = \{x_1, \ldots, x_N\}$ of N random samples containing one positive sample and $N - 1$ negative samples for latent representation c, and optimized with the contrastive loss $\mathcal{L}_N$. Then, the mutual information $\mathcal{M}(c, x_+)$ between c and its positive samples $x_+ \in \boldsymbol{X}$ has a lower bound*

$$\mathcal{M}(c, x_+) \geq \log(N) - \mathcal{L}_N. \tag{12}$$

The above lemma illustrates that when we optimize the contrastive learning loss $\mathcal{L}_N$, it enhances the lower bound of mutual information $\mathcal{M}(c, x_+)$. In other words, it reduces the representation difference between $x+$ and $c$. Since our FedGLCL has the same form with the $L_N$ of (Oord et al., 2018), the above lemma can be directly applied in our setting. The set $X$ can be considered as global text embeddings $\boldsymbol{T}$ and $c$ is replaced with local image embeddings $\boldsymbol{I}$. Therefore, we provide the following theorem:

**Theorem 1.** *Consider a FL system with K clients, and K training instances $\{(x_k, \sigma)_{k=1}^K\}$ from each client, satisfy $(x_k, \sigma) \sim \mathbb{P}_k$, where $\sigma \in [N]$ represents the class index. Let $\boldsymbol{T} = \{\boldsymbol{T}_1, \ldots, \boldsymbol{T_N}\}$ be a set of N text embeddings containing one positive sample and $N - 1$ negative samples, local image embedding $I_k = f(x_k; \boldsymbol{w}_k)$. Then, the total mutual information between K local image embeddings and their positive sample $\boldsymbol{T}_\sigma$ has a lower bound*

$$\sum_{k=1}^{K} \mathcal{M}(\boldsymbol{I}_k, \boldsymbol{T}_\sigma) \geq K \log(N) - \sum_{k=1}^{K} \mathcal{L}_k. \tag{13}$$

*Proof.* In FL, each local training is conducted independently and in parallel. Thus, for each client $k \in [K]$, we have

$$\mathcal{M}(\boldsymbol{I}_k, \boldsymbol{T}_\sigma) \geq \log(N) - \mathcal{L}_k \tag{14}$$

Therefore, with sufficient updates of local models, $\boldsymbol{I}_k$ converges to $\boldsymbol{T}_v$. The total mutual information of $K$ clients satisfy

$$\sum_{k=1}^{K} \mathcal{M}(\boldsymbol{I}_k, \boldsymbol{T}_\sigma) \geq (\log(N) - \mathcal{L}_1) + (\log(N) - \mathcal{L}_2) + \ldots + (\log(N) - \mathcal{L}_K)$$

$$= K \log(N) - \sum_{k=1}^{K} \mathcal{L}_k \tag{15}$$

Theorem 1 implies that minimizing the global empirical loss (Eq. (1)) is equivalent to maximizing the sum of mutual information between the feature representations of $K$ clients and the positive text embedding. Due to the independent training of each client, $\sum_{k=1}^{K} \mathcal{L}_k$ is minimized and $\sum_{k=1}^{K} \mathcal{M}(\boldsymbol{I}_k, \boldsymbol{T}_\sigma)$ is maximized only if $\forall \boldsymbol{I}_k \to \boldsymbol{T}_v$. Since $\boldsymbol{T}_\sigma$ is consistent and fixed for all clients, FedGLCL ensures that all local embeddings converge to the same target, thereby mitigating the feature difference. Notably, both the number of samples and the number of clients are $K$, so we use the same notation.

Furthermore, we offer additional insight into how our introduced loss function regulates local training, aiming to prevent overfitting on the majority class of local datasets.

**Theorem 2.** *In FedGLCL, the loss function $\mathcal{L}_k$ of client $K$ can be expressed as:*

$$\mathcal{L}_k = \ell^+(x_i; \boldsymbol{T}_{y_i}; \boldsymbol{w}_k) + \ell^-(x_i; \{\boldsymbol{T}_j\}_{j \in [N], j \neq y_i}; \boldsymbol{w}_k), \tag{16}$$

*where $\ell^+$ is the target objective, and $\ell^-$ is the regulation term to prevent overfitting.*

*Proof.* The local loss function can be transformed:

$$
\begin{aligned}
\mathcal{L}_k &= \mathbb{E}_{(x_i, y_i) \sim \mathbb{P}_k} \ell \\
&= \mathbb{E}_{(x_i, y_i) \sim \mathbb{P}_k} - \log \frac{\exp(\delta_{i, y_i}/\tau)}{\sum_{j \in [N], j \neq y_i} \exp(\delta_{i,j}/\tau)} \\
&= \mathbb{E}_{(x_i, y_i) \sim \mathbb{P}_k} \sum_{j \in [N], j \neq y_i} \delta_{i,j}/\tau - \delta_{i, y_i}/\tau \\
&= \mathbb{E}_{(x_i, y_i) \sim \mathbb{P}_k} \frac{1}{\tau} \sum_{j \in [N], j \neq y_i} \frac{\boldsymbol{I}_i \cdot \boldsymbol{T}_j^\top}{\|\boldsymbol{I}_i\|\|\boldsymbol{T}_j\|} - \frac{\boldsymbol{I}_i \cdot \boldsymbol{T}_{y_i}^\top}{\|\boldsymbol{I}_i\|\|\boldsymbol{T}_{y_i}\|} \\
&= \mathbb{E}_{(x_i, y_i) \sim \mathbb{P}_k} \frac{1}{\tau} \sum_{j \in [N], j \neq y_i} \frac{f(x_i; \boldsymbol{w}_k) \cdot \boldsymbol{T}_j^\top}{\|f(x_i; \boldsymbol{w}_k)\|\|\boldsymbol{T}_j\|} - \frac{f(x_i; \boldsymbol{w}_k) \cdot \boldsymbol{T}_{y_i}^\top}{\|f(x_i; \boldsymbol{w}_k)\|\|\boldsymbol{T}_{y_i}\|} \\
&= \ell^-(x_i; \{\boldsymbol{T}_j\}_{j \in [N], j \neq y_i}; \boldsymbol{w}_k) + \ell^+(x_i; \boldsymbol{T}_{y_i}; \boldsymbol{w}_k),
\end{aligned}
\tag{17}
$$

Then, we have

$$\ell^+(x_i; \boldsymbol{T}_{y_i}; \boldsymbol{w}_k) = \mathbb{E}_{(x_i, y_i) \sim \mathbb{P}_k} - \frac{1}{\tau} \frac{f(x_i; \boldsymbol{w}_k) \cdot \boldsymbol{T}_{y_i}^\top}{\|f(x_i; \boldsymbol{w}_k)\|\|\boldsymbol{T}_{y_i}\|} \tag{18}$$

$$\ell^-(x_i; \{\boldsymbol{T}_j\}_{j \in [N], j \neq y_i}; \boldsymbol{w}_k) = \mathbb{E}_{(x_i, y_i) \sim \mathbb{P}_k} \frac{1}{\tau} \sum_{j \in [N], j \neq y_i} \frac{f(x_i; \boldsymbol{w}_k) \cdot \boldsymbol{T}_j^\top}{\|f(x_i; \boldsymbol{w}_k)\|\|\boldsymbol{T}_j\|} \tag{19}$$

Compared with traditional cross-entropy loss, our local loss function introduces the information from all classes. By optimizing $\ell^+(x_i; \boldsymbol{T}_{y_i}; \boldsymbol{w}_k)$, $f(x_i; \boldsymbol{w}_k)$ is forced to approach $\boldsymbol{T}_{y_i}$. Besides, $\ell^-(x_i; \{\boldsymbol{T}_j\}_{j \in [N], j \neq y_i}; \boldsymbol{w}_k)$ can be considered as a regularization term to keep $f(x_i; \boldsymbol{w}_k)$ far away from $\forall \hat{\boldsymbol{T}} \in \{\boldsymbol{T}_j\}_{j \in [N], j \neq y_i}$, which can prevent the local model overfitting to majority class of local datasets.

## B.2 GENERALIZATION BOUNDS

In this section, we analyse the performance of FedGLCL from the perspective of generation error. Before giving our error bound, we provide the following lemma from (Tan et al., 2022b), which is a prototype-based FL framework.

**Lemma 2.** *Consider a FL system with $m$ clients. Let $D_1, D_2, \ldots, D_m$ be the true data distribution and $\hat{D}_1, \hat{D}_2, \ldots, \hat{D}_K$ be the empirical data distribution. Denote the projector head $h$ as the hypothesis from $\mathcal{H}$ and $d$ be the VC-dimension of $\mathcal{H}$. The total number of samples overall clients is $N$. Then, with probability at least $1 - \delta$,*

$$
\max_{(\theta_1, \theta_2, \ldots, \theta_m)} \left| \sum_{i=1}^{m} \frac{|D_i|}{N} L_{\mathcal{D}_i}\left(\theta_i; \mathbb{C}, \{\boldsymbol{C}_p\}_{p=1}^{m}\right) - \sum_{i=1}^{m} \frac{|D_i|}{N} L_{\hat{\mathcal{D}}_i}\left(\theta_i; \mathbb{C}, \{\boldsymbol{C}_p\}_{p=1}^{m}\right) \right|
$$
$$
\leq \sqrt{\frac{N}{2} \log \frac{(m+1)|\mathbb{C}|}{\delta}} + \sqrt{\frac{d}{N} \log \frac{eN}{d}},
\tag{20}
$$

*where $(m+1)\mathbb{C}$ is the total number of local and global prototypes, $e$ is the base of the natural logarithm.*

Since text embeddings $\boldsymbol{T}$ contain the class-wise information, we find the above lemma can be directly applied in our framework by considering $\boldsymbol{T}$ as a type of global prototype. This leads to the below theorem.

**Theorem 3.** *Let $\mathbb{P}_1, \mathbb{P}_2, \ldots, \mathbb{P}_K$ be the empirical data distribution and $\hat{\mathbb{P}}_1, \hat{\mathbb{P}}_2, \ldots, \hat{\mathbb{P}}_K$ be the true data distribution. Denote the projector $\mathcal{H}$ as the hypothesis from hypothesis space $\hat{\mathcal{H}}$ and $d$ be the VC-dimension of $\hat{\mathcal{H}}$. The number of text embeddings is $N$. With probability at least $1 - \delta$,*

$$
\max_{(\boldsymbol{w}_1, \boldsymbol{w}_2, \ldots, \boldsymbol{w}_K)} \left| \sum_{k=1}^{K} \gamma_k \mathcal{L}_{\mathbb{P}_k}(\boldsymbol{T}; \boldsymbol{w}_k) - \sum_{k=1}^{K} \gamma_k \mathcal{L}_{\hat{\mathbb{P}}_k}(\boldsymbol{T}; \boldsymbol{w}_k) \right|
$$
$$
\leq \sqrt{\frac{\sum_{k=1}^{K} n_k}{2} \log \frac{N}{\delta}} + \sqrt{\frac{d}{\sum_{k=1}^{K} n_k} \log \frac{e \sum_{k=1}^{K} n_k}{d}},
\tag{21}
$$

The above theorem indicates that FedGLCL can achieve the expected performance with an appropriate projection network and the number of classes.

## C DETAILS

### C.1 DATASET

We illustrate the detailed information of four datasets: CIFAR-100 and Fashion-MNIST in Table 5, Office-Caltech-10 in Table 6, and DomainNet in Table 7. For CIFAR-100 and Fashion-MNIST, the ratio of training and testing is provided by the dataset, where there are 50000 training instances for CIFAR-100 and 60000 training instances for Fashion-MNIST. Both of them have 10000 instances for testing. The training set will be split into multiple clients for training by two different Non-IID partition strategies (Diao et al., 2024; Zhang et al., 2022): 1) **Sharding**: we randomly allocate $\mathbb{C}$ classes to each client and keep an equal number of samples for each client. 2) **LDA**: we utilize the Latent Dirichlet Allocation (LDA) strategy to divide the data among clients, where each local dataset is sampled from a Dirichlet distribution ($\mathbb{P}_k \sim Dir(\beta)$). $\mathbb{C}$ and $\beta$ are hyper-parameters that control the data heterogeneity of experiments, and the smaller value indicates higher data heterogeneity. After training, we evaluate the global model of all methods on the testing set (Diao et al., 2024; Li et al., 2021). For Office-Caltech-10 and DomainNet, following (Li et al., 2020c; Zhou & Konukoglu, 2023), we partition the dataset of each client into `8:2` for `training:testing`. We report the test accuracy of each client and their average result for all methods.

### C.2 METHOD

It's worth noting that there are some important hyper-parameters in the compared baselines, which can affect the performance of the method. For these hyper-parameters and experimental details,

Table 5: **Detailed information of CIFAR-100** (Krizhevsky et al., 2009), **Fashion-MNIST** (Xiao et al., 2017), and **Tiny-ImageNet**.

| Property | CIFAR-100 | Fashion-MNIST | Tiny-ImageNet |
|---|---|---|---|
| # of train samples | 50000 | 60000 | 100000 |
| # of test samples | 10000 | 10000 | 10000 |
| # of classes | 100 | 10 | 200 |
| Image size | (32, 32, 3) | (28, 28, 1) | (64, 64, 3) |

Table 6: **Detailed information of Office-Caltech-10** (Gong et al., 2012). There are 4 clients in total.

| Property | Office-Caltech-10 | | | |
|---|---|---|---|---|
| | Amazon | Caltech | DSLR | Webcam |
| # of train samples | 766 | 898 | 125 | 236 |
| # of test samples | 192 | 225 | 32 | 59 |
| # of classes | 10 | 10 | 10 | 10 |
| Image size | (256, 256, 3) | (256, 256, 3) | (256, 256, 3) | (256, 256, 3) |

Table 7: **Detailed information of DomainNet** (Peng et al., 2019). There are 6 clients in total.

| Property | DomainNet | | | | | |
|---|---|---|---|---|---|---|
| | Clipart | Infograph | Painting | Quickdraw | Real | Sketch |
| # of train samples | 2103 | 2626 | 2472 | 4000 | 4867 | 2213 |
| # of test samples | 526 | 657 | 619 | 1000 | 1217 | 554 |
| # of classes | 10 | 10 | 10 | 10 | 10 | 10 |
| Image size | (256, 256, 3) | (256, 256, 3) | (256, 256, 3) | (256, 256, 3) | (256, 256, 3) | (256, 256, 3) |

Table 8: **Detailed structure of the projector for ResNet-50.** The projector is a lightweight module, consisting of two fully connected layers (FC). We list parameters with a sequence of input and output dimensions.

| Layer | Details | | |
|---|---|---|---|
| Layer1 | FC(2048, 2048), | ReLU, | Normalize |
| Layer2 | FC(2048, 640), | ReLU, | Normalize |

we strictly adhere to the descriptions provided by their paper. For example, the local objective function of FedProx, MOON and FedProto can be expressed as $\mathcal{L} = \mathcal{L}_1 + \mu\mathcal{L}_2$, where $\mathcal{L}_1$ and $\mathcal{L}_2$ are supervised loss and auxiliary Loss, $\mu$ is the hyper-parameter to control the influence of auxiliary Loss. For FedProx, we tune $\mu$ from the range {0.0001, 0.001, 0.01, 0.1}, and empirically set to 0.001 for CIFAR-100 and Fashion-MNIST, and 0.0001 for Office-Caltech-10 and DomainNet. For MOON and FedProto, the $\mu$ is set to 1, and the temperature parameter of MOON is set to 0.5. As for FedConcat, the encoder and classifier are trained independently, with the encoder being trained first, followed by the training of the classifier. The training communication rounds for the encoder are 50 rounds, while for the classifier, they are 1000 rounds. To ensure a fair comparison, all baselines employ the same model structure, integrating the projector into ResNet-50 (backbone, projector, classifier). The projector is a two-layer Multi-Layer Perceptron (MLP), which is presented in Table 8. However, FedGLCL differs by removing the classifier, retaining only the backbone and projector components. In the case of AlexNet, its classifier comprises three consecutive linear layers. FedGLCL discards the last linear layer of the classifier and utilises the remaining two linear layers as the projector while all baselines maintain the full classifier structure.

## D ADDITIONAL EXPERIMENTS

### D.1 PROMPTING ENGINEERING

To explore the impact of prompting engineering on the method, we build up a variant $M_1$ by removing prompt templates and directly using class names "{class}". The results in Table 9 show a varying degree of performance decline across the three datasets after removing the prompts. This

Table 9: **Ablation studies** about the prompting engineering of FedGLCL on three benchmarks. The $\beta$ is 0.5 for CIFAR-100.

| Variant | CIFAR-100 | Office-Caltech-10 | DomainNet |
|---|---|---|---|
| FedAvg | 66.67 | 60.25 | 66.28 |
| **FedGLCL** + "{class}" | 72.42 | 75.91 | 75.29 |
| **FedGLCL** + Prompt ensemble | 74.30 | 77.92 | 77.02 |
| **FedGLCL** + "a photo of {class}" | **73.52** | **77.35** | **76.54** |

Table 10: **Test accuracy (%)** with different text embeddings on three benchmarks. The $\beta$ is 0.5 for CIFAR-100.

| Variant | CIFAR-100 | Office-Caltech-10 | DomainNet |
|---|---|---|---|
| Random initialization tensors | 38.99 | 35.09 | 42.13 |
| Random orthogonal tensors | 67.58 | 62.60 | 69.22 |
| Proxy text embeddings | 72.80 | **77.52** | 75.83 |
| **FedGLCL** | **73.52** | 77.35 | **76.54** |

Table 11: **Test accuracy (%)** with vgg-11 on CIFAR-100. The $\beta$ is 0.5.

| FedAvg | FedProx | FedNova | Scaffold | MOON | FedProto | FedLC | FedConcat | FedGLCL |
|---|---|---|---|---|---|---|---|---|
| 57.57 | 54.64 | 58.12 | 58.43 | 60.12 | 59.63 | 55.52 | 60.94 | **65.45** |

Table 12: **Test accuracy (%)** with vgg-11 on Office-Caltech-10.

| FedAvg | FedProx | FedNova | Scaffold | MOON | FedProto | FedBN | FedFA | FedGLCL |
|---|---|---|---|---|---|---|---|---|
| 74.97 | 73.54 | 73.27 | 72.44 | 75.05 | 78.64 | 81.55 | 82.68 | **85.30** |

suggests that text prompts contribute to enhancing the semantic information of text embeddings. However, even after removing textual prompts, FedGLCL still significantly outperforms traditional label-driven representation learning methods, demonstrating the superiority of language-driven representation learning for heterogeneous data.

Besides, following previous works (Zhou et al., 2022), we employ prompt ensemble by feeding prompt-engineered texts into the text encoder with 85 different prompt templates, and average 85 text embeddings of the same class. The results show that this ensemble strategy yields better text embeddings, leading to improved performance. This further demonstrates the scalability of FedGLCL.

## D.2 DIFFERENT TEXT EMBEDDINGS

In this section, we explore the impact of different types of text embeddings on our approach. First, we conduct experiments by replacing the global text embedding with randomly initialized and randomly orthogonal tensors implemented by Pytorch. The main idea of FedGLCL is to leverage the semantic information contained in global text embeddings for global supervision. However, as shown in Table 10, randomly initialized feature embeddings struggle to provide effective semantic supervision, leading to a sharp decline in performance. Random orthogonal matrices can be a type of supervision, but manually crafted feature embeddings significantly lower the quality of global text embeddings, which are learned from massive data. In addition, manually crafted feature embeddings can hinder the convergence of the model.

Besides, we use fake class names as proxies by swapping the class names between Office-Caltech-10 and DomainNet, and randomly selecting 100 class names from ImageNet (Deng et al., 2009) for CIFAR-100. Then, we utilise the text encoder to encode them as the global text embedding for both training and inference. Surprisingly, we found that it achieved comparable performance to using text embeddings from real classes as shown in Table 10. In FedGLCL, local image encoders are trained from scratch without any prior knowledge. Therefore, the critical factor is the consistency of text embeddings between training and testing. This ensures that the encoded text embeddings can effectively guide the learning process, regardless of whether the class names perfectly correspond to the images. While the proxy class names are not the real classes of images, they are still in the domain covered by the pre-trained text encoder. Consequently, the global text embeddings still provide

Table 13: **Test accuracy (%)** with different text encoders on three benchmarks. The $\beta$ is 0.5 for CIFAR-100.

| Variant | CIFAR-100 | Office-Caltech-10 | DomainNet |
|---|---|---|---|
| Glove | 68.95 | 71.51 | 70.48 |
| CLIP/ResNet-50 | 73.30 | 77.26 | 76.41 |
| CLIP/ViTB-16 | 72.88 | 75.99 | 76.09 |
| **Bert-base** (default) | **73.52** | **77.35** | **76.54** |

Table 14: **Test accuracy (%)** with ResNet-50 on CIFAR-100-LT (Shang et al., 2022).

| FedAvg | FedProx | FedNova | Scaffold | MOON | FedProto | FedLC | FedConcat | FedGLCL |
|---|---|---|---|---|---|---|---|---|
| 45.29 | 48.23 | 45.49 | 49.40 | 51.13 | 50.76 | 48.92 | 50.53 | **56.14** |

Table 15: **Training efficiency** with ResNet-50 on CIFAR-100.

| Method | FedAvg | FedProx | FedNova | Scaffold | MOON | FedProto | FedLC | FedGLCL |
|---|---|---|---|---|---|---|---|---|
| Time/Round (s) | 12.58 | 21.69 | 13.68 | 18.83 | 21.14 | 15.05 | 13.13 | **14.13** |
| Rounds | 88 | 91 | 93 | 96 | 93 | 87 | 79 | **92** |
| Total Time (s) | 1107.04 | 1973.79 | 1272.24 | 1807.68 | 1966.02 | 1309.35 | 1037.27 | **1299.96** |

meaningful semantic supervision for training. This result highlights that the success of language supervision is not dependent on the precise semantic relationships between class names and images but rather on the overall structure and knowledge encoded in the global text embeddings. This not only offers effective supervision for training local image encoders but, more importantly, ensures the alignment of all local image feature spaces with the global text feature space for mitigating drift in local models. Besides, this also suggests that we can use proxy texts to deal with the domain shift of text and privacy encryption.

## D.3 DIFFERENT IMAGE BACKBONES

We conducted experiments using vgg-11 (Simonyan & Zisserman, 2014) for all methods on CIFAR-100 and Office-Caltech-10, where the $\beta$ is set to 0.5 for CIFAR-100. The results are as shown in Table 11 and 12. As one can see, after employing different network architectures, the improvement of FedGLCL compared to FedAvg remains notable, *i.e.*, **7.88**% and **10.33**% on the two datasets, respectively. Besides, FedGLCL consistently outperforms all compared baselines under two different non-IID settings. The above results demonstrate the generalizability of FedGLCL across different network architectures

## D.4 DIFFERENT TEXT ENCODERS

To explore the impact of different pre-trained language models, we used another popular vision-language model, CLIP (Radford et al., 2021). Specifically, we only use the text encoder of CLIP/ResNet-50 and CLIP/ViTB-16 as the text encoder of FedGLCL, and the results are presented in Table 13. After using different text encoders, our method still outperformed the best baseline. This demonstrates the robustness of FedGLCL to various text encoders.

Besides, we compared Glove (Pennington et al., 2014), a traditional word embedding method. The results show that GloVe outperforms FedAvg, but achieves lower performance than Bert and CLIP. This is because Bert and CLIP employ deep neural networks to learn data representations through supervised learning on large-scale datasets, allowing them to capture semantic information more effectively.

## D.5 EXPERIMENT ON LONG-TAILED SETTING

To further validate the effectiveness of FedGLCL in mitigating overfitting to the major classes, following previous work (Shang et al., 2022), we conducted long-tailed FL experiments on CIFAR-100-LT. The long-tailed distribution is controlled by imbalance factors (IF), which is set to 10, and $\beta$ is set to 0.5. The results, as illustrated in Table 14, show that the improvement of FedGLCL over

Table 16: **Test accuracy (%)** with ResNet-50 on Tiny-ImageNet. The $\beta$ is 0.5.

| FedAvg | FedProx | FedNova | Scaffold | MOON | FedProto | FedLC | FedConcat | FedGLCL |
|--------|---------|---------|----------|-------|----------|-------|-----------|---------|
| 47.54 | 46.61 | 47.11 | 48.09 | 48.97 | 48.16 | 45.97 | 49.24 | **51.05** |

Table 17: **Results of statistical significance**. For accuracy, we report mean $\pm$ std over three trials.

| Method | CIFAR-100 | | Office-Caltech-10 | | DomainNet | |
|--------|-----------|---------|-------------------|---------|-----------|---------|
| | Accuracy | P-value | Accuracy | P-value | Accuracy | P-value |
| FedAvg | $66.64 \pm 0.14$ | 0.0016 | $60.57 \pm 0.85$ | 0.0003 | $66.58 \pm 0.70$ | 0.0007 |
| FedConcat | $71.09 \pm 0.51$ | 0.0138 | - | - | - | - |
| FedFA | - | - | $71.50 \pm 1.03$ | 0.0026 | $72.27 \pm 0.72$ | 0.0031 |
| **FedGLCL** | $\mathbf{74.10 \pm 0.41}$ | - | $\mathbf{77.84 \pm 0.96}$ | - | $\mathbf{77.13 \pm 0.53}$ | - |

Table 18: **Test accuracy (%)** with AlexNet on Office-Caltech-10 (Gong et al., 2012), where the image backbone is initialized with pre-trained weights from ImageNet.

| FedAvg | FedProx | FedNova | Scaffold | MOON | FedProto | FedBN | FedFA | FedPCL | FedGLCL |
|--------|---------|---------|----------|-------|----------|-------|-------|--------|---------|
| 64.99 | 62.46 | 65.50 | 61.33 | 66.78 | 69.37 | 71.87 | 74.91 | 72.46 | **81.81** |

FedAvg is more significant in long-tailed FL settings as large as **10.85**% (higher than **6.85**% in Table 2). This further demonstrates that FedGLCL can prevent overfitting to the major classes.

### D.6 TRAINING EFFICIENCY

We present the local training time per round of single client on CIFAR-100, and the average result of all clients is reported. We also report the number of rounds for each method to reach the best accuracy. Notably, since FedConcat's backbone and classifier are trained independently, the number of rounds and training time for each stage differ, so it is a challenge to evaluate its time using the above strategy. The results are presented in Table 15. Compared to FedAvg, the additional training overhead in FedGLCL primarily comes from the similarity computation between image and text features. However, this overhead is minimal as it leverages PyTorch's efficient matrix operations, resulting in only a slight increase in Time/Round. Additionally, we compared FedGLCL with other methods that incorporate auxiliary losses, such as FedProx, MOON, and FedProto. The auxiliary losses in FedProx and MOON lead to significantly higher overhead compared to our method. In summary, both the training time per round and the total training time for FedGLCL are acceptable.

### D.7 EXPERIMENTS ON LARGER-SCALE DATASETS

We conduct experiments using a larger-scale dataset, Tiny-ImageNet[2], and its detailed information is presented in Table 5. We use the same experimental setup as for CIFAR-100, and $\beta$ is set to 0.5. The results in Table 16 show that FedGLCL consistently achieves the best performance compared to all baselines, further demonstrating the effectiveness of our method.

### D.8 STATISTICAL SIGNIFICANCE

In this section, we conduct two additional trials using different random seeds and report the mean and standard deviation (std) across all three trials. Furthermore, to evaluate the statistical significance of the performance improvements, we perform a paired t-test between the baseline and our method, reporting the corresponding p-value. The above results of FedAvg, FedGLCL, and two best baselines, *i.e.*, FedConcat and FedFA, are presented in Table 17. It can be observed that the p-values for all baselines are less than 0.05, indicating the statistical significance of the performance improvements achieved by our method.

---

[2]https://www.kaggle.com/c/tiny-imagenet/overview

Table 19: **Test accuracy (%) on ProstateMRI** (Liu et al., 2020). For a detailed comparison, we present the test accuracy of each client. **Avg.** denotes the average accuracy of all clients

| Method | ProstateMRI | | | | | | |
|---|---|---|---|---|---|---|---|
| | BIDMC | HK | I2CVB | BMC | RUNMC | UCL | Avg. |
| FedAvg | 87.66 | 94.48 | **96.00** | 90.46 | 93.21 | 87.47 | 91.55 |
| **FedGLCL** | **91.81** | **94.89** | 95.95 | **92.08** | **93.34** | **90.68** | **93.12** |

### D.9 EXPERIMENTS WITH PRE-TRAINED IMAGE MODELS

We conducted additional experiments on the Office-Caltech-10 dataset using a pre-trained image backbone. In this experiment, the image backbone (AlexNet) was initialized with pre-trained weights from ImageNet, while all other settings remained unchanged. The results in Table 18 demonstrate that incorporating the pre-trained image backbone improved the performance of all methods, with our approach consistently achieving the best results. This confirms that under fair comparisons (where all methods either utilize the pre-trained image backbone or do not), FedGLCL significantly outperforms other methods, further highlighting its effectiveness and superiority.

Besides, we compared a new baseline, *i.e.*, FedPCL (Tan et al., 2022b), a prototype-based FL method specifically designed for pre-trained image backbones. As we can see, FedPCL performs significantly worse than our method, demonstrating the superiority of FedGLCL over prototype-based FL methods.

### D.10 EXPLORATION ON MORE TASKS

In this section, we extend FedGLCL to more visual tasks. Inspired by (Li et al., 2022a), we extend the image-text alignment to pixel-text alignment for image segmentation. We conducted experiments on ProstateMRI (Liu et al., 2020), a widely used federated medical image segmentation benchmark. This benchmark has six clients including BIDMC, HK, I2CVB, BMC, RUNMC, and UCL, each from a different domain. We used U-Net as the segmentation network and aligned the output of the final layer with the text features. Since this benchmark involves a binary segmentation task, we use "foreground" and "background" as the class names. The communication rounds are set to 200, with local rounds set to 1. The results are presented in Table 19. It can be observed that FedGLCL achieves certain improvements compared to FedAvg, indicating that our method has the potential to be applied to other tasks to address non-IID data issue.

## E PRIVACY SECURITY

FedGLCL effectively utilizes language-driven representation without leaking privacy. (1) We share the encoded text embeddings instead of directly sharing the category text. Additionally, we conduct the text encoding process on the server side, making it invisible to individual clients, thus preventing reverse inference of the text. (2) Furthermore, the introduced prompts can increase the complexity of the text, thereby further enhancing privacy security. (3) Finally, we can further enhance privacy security by using proxy texts.

## F LIMITATION

Domain shift in pre-trained models is a fundamental challenge and can be mitigated in this work. Actually, FedGLCL is not limited by domain shift issues because language exhibits consistency in real-world applications. For example, the word "cat" holds the same semantics in both natural images and cartoon image tasks. However, for image backbones, the natural images and the cartoon images hold two entirely different data distributions. On the contrary, the image encoder of

FedGLCL is trained from scratch without any prior knowledge and thus is irrelevant to the pre-trained CLIP image encoder. For the domains not covered by text encoder, we can utilize the corresponding text encoder for specific data domains, *e.g.*, BioBERT (Lee et al., 2020) for medical image tasks. Besides, the ablation results in Table 10 indicate that we can use in-domain class names as proxies when deployed in domains not well-covered by a pre-trained text encoder. This can broaden the application scenarios of FedGLCL. Although FedGLCL is not limited by domain shift, it may have limitations in larger vocabulary scenarios (*e.g.*, 10,000 target categories). Text encoders may struggle to effectively encode rich class vocabulary to provide effective semantic information. In the future work, we will thoroughly explore it.

