# OpenReview forum: "On the Importance of Language-driven Representation Learning for Heterogeneous Federated Learning"
_ICLR.cc/2025/Conference — ICLR 2025 Poster_

### Official Review · Reviewer_Q1wc · 2024-10-30

**Soundness:** 3
**Presentation:** 2
**Contribution:** 2
**Rating:** 6
**Confidence:** 3

**Summary:**

This paper solves the heterogeneous federated learning (in the image classification field) via a pre-trained and fixed text encoder (BERT) on the server and learned image encoders on the clients. The classification is done by clip-style image-text embedding similarity comparison. The authors report superior performance over traditional heterogeneous federated learning methods (FedAvg, etc.), and provide scalability analysis, efficient analysis, and ablation study.

**Strengths:**

1. The proposed method shows superior performance.
2. Comprehensive analysis is provided.
3. The usage of Bert embedding as a prototype vector for FL image classification is novel.

**Weaknesses:**

1. There are few comparisons with prototype FL (like [1] and [2]).
2. The method proposed has some limitations. It is restricted to image classification and requires a pre-trained model that can give good embeddings of the classification labels. This seems to exclude many FL scenarios. (I am not to say that using Office-Caltech-10 and DomainNet is not enough. I just wonder if it may not be as versatile as FL methods like FedAvg. Yet I still appreciate this paper as a good method for image classfication.)


[1] Federated Learning from Pre-Trained Models: A Contrastive Learning Approach https://proceedings.neurips.cc/paper_files/paper/2022/file/7aa320d2b4b8f6400b18f6f77b6c1535-Paper-Conference.pdf
[2] FedProto: Federated Prototype Learning across Heterogeneous Clients https://arxiv.org/pdf/2105.00243

**Questions:**

1. Comparison/Detailed discussion about prototype FL
2. Can it be adjusted for tasks besides image classification?
3. Can traditional word embedding such as Glove also achieve a performance comparable to Bert's? (just curious about this)

---

> ### Author Response · Authors · 2024-11-20
> **Official Comment by Authors**
>
> We thank the reviewer for his/her constructive comments and provide our point-wise replies as follows.
>
> > **Q1:** Comparison with prototype FL.
>
> We have compared with FedProto [2] in our experiments, and the results are presented in Tables 1 and 2. FedPCL [1] uses the pre-trained image backbone. For a fair comparison, we conducted additional experiments using the pre-trained image backbone (AlexNet) on the Office-Caltech-10. Specifically, we initialized the image backbone with pre-trained weights from ImageNet and kept all other settings unchanged. As we can see, FedPCL performs significantly worse than our method, demonstrating the superiority of FedGLCL over prototype-based FL methods. The results have been included in the revision (Appendix D.9 Table 18)
>
>
> |Method |FedAvg|FedProx |FedNova|Scaffold| MOON| FedProto| FedBN |FedFA| FedPCL |FedGLCL |
> |:-----:|:----:|:-----:|:----:|:-----:|:----:|:-----:|:----:|:-----:|:----:|:----:|
> |Office-Caltech-10| 64.99 | 62.46 | 65.50 | 61.33 | 66.78 | 69.37 | 71.87 |74.91| 72.46 | 81.81 |
>
>
> > **Q2:** More tasks.
>
> As stated in Line 052, this work primarily uses visual recognition as an example, but it can be extended to more visual understanding tasks, such as image segmentation. Inspired by [3], we can extend the image-text alignment to pixel-text alignment for image segmentation. We conducted experiments on ProstateMRI [4], a widely used federated medical image segmentation benchmark. This benchmark has six clients including BIDMC, HK, I2CVB, BMC, RUNMC, and UCL, each from a different domain. We used U-Net as the segmentation network and aligned the output of the final layer with the text features. Since this benchmark involves a binary segmentation task, we use "foreground" and "background" as the class names. The communication rounds are set to 200, with local rounds set to 1. The results are presented below. It can be observed that FedGLCL achieves certain improvements compared to FedAvg, indicating that our method has the potential to be applied to other tasks to address non-IID data issues.
>
> |Method|BIDMC|HK|I2CVB|BMC|RUNMC|UCL|Avg.|
> | :-----|:----: |:----: |:----:| :-----|:----: |:----: |:----:|
> |FedAvg| 87.66|94.48|96.00|90.46|93.21|87.47|91.55|
> FedGLCL|91.81|94.89|95.95|92.08|93.34|90.68|93.12|
>
>
> > **Q3:** Curious about the peformance of Glove.
>
> Thanks for your suggestion. We conducted additional experiments using GloVe to encode global text embeddings. The results, presented below, show that GloVe outperforms FedAvg, but achieves lower performance than Bert and CLIP. This is because Bert and CLIP employ deep neural networks to learn data representations through supervised learning on large-scale datasets, allowing them to capture semantic information more effectively. The results have been included in the revision (Appendix D.4 Table 13)
>
> |Text Encoder | CIFAR-100 | Office-Caltech-10 | DomainNet |
> | :-----|:----: |:----: |:----:|
> |FedAvg| 66.67| 60.25 |66.28 |
> | Glove | 68.95 | 71.51 | 70.48|
> |CLIP/ResNet-50| 73.30 | 77.26 | 76.41|
> |CLIP/ViTB-16| 72.88 | 75.99| 76.09|
> |Bert-base (default)| 73.52 | 77.35| 76.54 |
>
> **References**
>
> [1] Tan, Yue, et al. "Federated learning from pre-trained models: A contrastive learning approach." Advances in neural information processing systems 35 (2022): 19332-19344.
>
> [2] Tan, Yue, et al. "Fedproto: Federated prototype learning across heterogeneous clients." Proceedings of the AAAI Conference on Artificial Intelligence. Vol. 36. No. 8. 2022.
>
> [3] Li, Boyi, et al. "Language-driven Semantic Segmentation." International Conference on Learning Representations. 2022.
>
> [4] Liu, Quande, et al. "MS-Net: multi-site network for improving prostate segmentation with heterogeneous MRI data." IEEE
> transactions on medical imaging 39.9 (2020): 2713-2724.

---

> > ### Comment · Reviewer_Q1wc · 2024-11-24
> >
> > Thanks for your response. I raise my rating to 6.

---

> > > ### Author Response · Authors · 2024-11-24
> > > **Official Comment by Authors**
> > >
> > > Thanks for your valuable time to respond to our feedback. We are very happy to see that your concerns have been fully addressed.

---

### Official Review · Reviewer_4piz · 2024-11-01

**Soundness:** 3
**Presentation:** 3
**Contribution:** 3
**Rating:** 6
**Confidence:** 3

**Summary:**

The paper presents FedGLCL, a novel federated learning (FL) framework designed to address the challenges posed by non-Independent and Identically Distributed (Non-IID) data. The framework leverages a pretrained VLM, CLIP, specifically through contrastive learning that integrates global language and local image features. The proposed method aims to harmonize feature learning across different clients, reducing variance in local representations and mitigating overfitting to majority classes. The paper includes extensive theoretical and empirical analyses, demonstrating the superiority of FedGLCL over state-of-the-art FL algorithms in various non-IID scenarios.

**Strengths:**

1. The utilization of large pretrained CLIP into FL is a unique and innovative direction. By leveraging contrastive learning to align local image features with global textual data, the paper proposes a novel solution to tackle the Non-IID problem.
2. The paper provides a solid theoretical foundation for FedGLCL, explaining how the framework can mitigate issues related to Non-IID data. This theoretical backing strengthens the credibility of the proposed method.
3.The authors conduct a comprehensive set of experiments, comparing FedGLCL with multiple state-of-the-art FL algorithms across different non-IID scenarios. The results demonstrate significant improvements, validating the effectiveness of the proposed framework.

**Weaknesses:**

1. Clarity and presentation issue: It is not clear which pretrained vision language models are used in the experiments. CLIP? Align? BLIP?

2. It is better to compare the performance of different text encoders from various pretrained VLMs, because the proposed method heavily relies on the pretrained text encoder.

**Questions:**

Are the datasets used in the experiments part of the pretraining of VLMs? If this is the case, then your framework achieves better performance is due to the pretraining stage.

---

> ### Author Response · Authors · 2024-11-20
> **Official Comment by Authors**
>
> We thank the reviewer for his/her constructive comments and provide our point-wise replies as follows.
>
> > **Q1:** Which pretrained vision language models are used in the experiments?
>
> In our default setting (Lines 363-372), we used a language model, Bert-base, as the text encoder rather than a vision-language model. Besides, the image backbone is trained from scratch without loading any pre-trained models.
>
>
> > **Q2:** Comparison on different text encoders.
>
> |Text Encoder | CIFAR-100 | Office-Caltech-10 | DomainNet |
> | :-----|:----: |:----: |:----:|
> | Best baseline | 71.82 | 71.36 | 72.94|
> |CLIP/ResNet-50| 73.30 | 77.26 | 76.41
> |CLIP/ViTB-16| 72.88 | 75.99| 76.09|
> |Bert-base (default)| 73.52 | 77.35| 76.54 |
>
> Thanks for your comments. We have already presented the results of this experiment in Appendix D.4 (Table 13). As shown, our method still outperformed the best baseline after using different text encoders. This demonstrates the robustness of FedGLCL to various text encoders.
>
> > **Q3:** Are the datasets used in the experiments part of the pre-training of VLMs?
>
> The datasets used in the experiments are **NOT** used for pre-training. The used text encoder, Bert-base, is trained on the pure language dataset and does **NOT** contain any knowledge from four used datasets (image datasets). Besides, the image backbone is trained from scratch without loading any pre-trained models. This indicates that the improvement in our method stems not from the pre-training stage, but from our novel learning mechanism.

---

### Official Review · Reviewer_bW7e · 2024-11-04

**Soundness:** 3
**Presentation:** 3
**Contribution:** 3
**Rating:** 6
**Confidence:** 3

**Summary:**

This paper proposes a novel approach, FedGLCL, to learn image models in non-IID federated settings. Their approach leverages a fixed frozen set of class embeddings (from a language model) and locally updates image representations in a CLIP-like contrastive manner. When tested on multiple variations of non-IID federated learning benchmarks (e.g. significant label shift/imbalance), this approach is shown to notably outperform other training/aggregation-based interventions without introducing additional computation/communication costs. Furthermore, the authors provide several additional ablations and theoretical analyses to deepen understanding of their results.

**Strengths:**

**Originality and Significance.** Federated learning with non-IID client data is a well-motivated/challenging problem and the authors' approach, while simple, is novelly and effectively applied in this setting. As mentioned in Sec 5.4, this approach also importantly does not come with any decreases in efficiency.

**Quality.**  Overall, the results for their method are quite strong compared to a variety of baselines and on several variations of the evaluations. The authors also did a great job ablating aspects of their method (particularly in the Appendix), addressing several of the questions that came up while I was reading the paper. In particular, I appreciated the ablations on the text embeddings, which show that it is important to both (1) keep the embeddings fixed; (2) use a pre-trained language model instead of custom class embeddings. These help establish that contrastive-style training itself is not the main explainer for good performance.

**Clarity.** Generally, l found the paper to be well-written and easy to follow. Particularly, the authors did a good job contextualizing their methods/results with previous works.

**Weaknesses:**

**More discussion about the impact of language supervision.** One result I found particularly curious (and which I believe should be highlighted more in the main paper) was that using embeddings for randomly mismatched class names results in comparable performance to using embeddings for the correct class names. Before seeing this result, I would have thought that the semantic information and relationships between classes was being usefully leveraged (e.g., similar classes being closer together in text embedding space). But based on this result, I'm not sure then to what extent the "broad linguistic knowledge" emphasized in the abstract is important in explaining FedGLCL's success? Perhaps the authors can expand upon their discussion of this in the Appendix and comment more.

**Measuring the impact of FedGLCL-style training in centralized/IID-Federated settings** It would be interesting to see how well the contrastive style training from FedGLCL performs in the non-federated setting (and perhaps also the IID federated setting). This would shed light on whether the gains in the non-IID federated setting might also be explained by the contrastive learning approach in FedGLCL-being more effective generally on these tasks v.s. if it is uniquely helpful for overcoming challenges in non-IID federated settings. Relatedly, it would also be nice to discuss your results in relation to Fang et al., 2022 (https://arxiv.org/abs/2205.01397), which included results assessing the impact of language supervision on CLIP pre-training. in their case, they find that it does not have a major impact on the final robustness of CLIP models.

**Standard errors for results.** It would be nice if the empirical results came with standard errors. That being said, I can acknowledge that the current gaps between the proposed method and previous baselines seem to be large/consistent across evaluation settings and that repeated runs might be costly.

**Questions:**

Most questions I had were covered in the Weaknesses section. The only other question I had was if you considered any settings involving pretrained image models. On the surface, this might better reflect practical deployments for image-based tasks?

---

> ### Author Response · Authors · 2024-11-20
> **Official Comment by Authors**
>
> We thank the reviewer for his/her constructive comments and provide our point-wise replies as follows.
>
> > **Q1:** Discussion about the impact of language supervision.
>
> Sorry for the confusion. We clarify it as follows and make it clear in the  revision (Line 1069-1079).
>
> (1) **Impact of language supervision:** Non-IID data leads to significant drift in local models. The key idea of FedGLCL is to align the local image feature space with the global text feature space, which benefits learning a consistent feature representation and prevents overfitting to the majority classes across different local models.
>
> (2) **Explanation of experimental results:** In FedGLCL, local image encoders are trained from scratch without any prior knowledge. Therefore, the critical factor is the consistency of text embeddings between training and testing. This ensures that the encoded text embeddings can effectively guide the learning process, regardless of whether the class names perfectly correspond to the images. While the proxy class names are not the real classes of images, they are still in the domain covered by the pre-trained text encoder. Consequently, the global text embeddings still provide meaningful semantic supervision for training.
>
> (3) **Success of linguistic knowledge:** This result highlights that the success of language supervision is not dependent on the precise semantic relationships between class names and images but rather on the overall structure and knowledge encoded in the global text embeddings. This not only offers effective supervision for training local image encoders but, more importantly, ensures the alignment of all local image feature spaces with the global text feature space for mitigating drift in local models.
>
>
>
> > **Q2:** Measuring the impact of FedGLCL-style training in centralized setting.
>
> (1) We present the results of FedAvg and FedGLCL on Office-Caltech-10 under both centralized and federated settings. The results show that FedGLCL achieves significantly greater performance improvements over FedAvg in the federated setting compared to the centralized setting. This demonstrates that the improvements of FedGLCL primarily arise from its ability to address the non-IID problem rather than solely from the contrastive learning approach.
>
> |Method| FedAvg (Centralized) | FedGLCL (Centralized)| FedAvg (Federated) |FedGLCL (Federated) |
> | :-----|:----: |:-----:|:----: |:----: |
> |Office-Caltech-10|83.32|85.55 | 60.25| 77.35 |
>
> (2) This paper investigates the issue of non-IID data across different clients in FL, which leads to significant drift in local models. In contrast, reference [1] explores the distributional differences between pre-training data and downstream task data. They are entirely two different problems. Therefore, our results cannot be directly compared or discussed in relation to the results of reference [1].
>
>
>
> > **Q3:** Standard errors for results.
>
> Thanks for recognizing the **large** improvement of our method. We conduct two additional trials using different random seeds and report the mean and standard deviation (std) across all three trials (mean $\pm$ std). Furthermore, to evaluate the statistical significance of the performance improvements, we perform a paired t-test between the baseline and our method, reporting the corresponding p-value. The above results of FedAvg, FedGLCL, and two best baselines, i.e., FedConcat and FedFA, are presented below. It can be observed that the p-values for all baselines are **less than 0.05**, indicating the statistical significance of the performance improvements achieved by our method. These results have been included in the revision (Appendix D.8 Table 17).
>
>
> |Text Encoder | CIFAR-100| p-value | Office-Caltech-10 | p-value| DomainNet | p-value |
> | :-----|:----: |:----: |:----:|:----:|:----:|:----:|
> |FedAvg| 66.64 $\pm$ 0.14| 0.0016 | 60.57 $\pm$ 0.85 | 0.0003 | 66.58 $\pm$ 0.70 | 0.0007|
> |FedConcat| 71.09 $\pm$ 0.51 | 0.0138| - | - | -| - |
> |FedFA| -| - |71.50 $\pm$ 1.03 | 0.0026 | 72.27 $\pm$ 0.72 | 0.0031 |
> |FedGLCL | 74.10$\pm$ 0.41 |-| 77.84 $\pm$ 0.96 | -| 77.13 $\pm$ 0.53| - |

---

> ### Author Response · Authors · 2024-11-20
> **Official Comment by Authors**
>
> > **Q4:** Add experiments with pre-trained image models.
>
> Thanks for your suggestion. We conducted additional experiments using pre-trained image models on the Office-Caltech-10. Specifically, we initialized the image backbone (AlexNet) with pre-trained weights from ImageNet and kept all other settings unchanged. The results show that incorporating the pre-trained image backbone improved the performance of all methods, and our approach consistently achieved the best results. This confirms that under fair comparisons (where all methods either utilize the pre-trained image backbone or do not), FedGLCL significantly outperforms other methods, further demonstrating its effectiveness and superiority. The results have been included in the revision (Appendix D.9 Table 18).
>
> |Method |FedAvg|FedProx |FedNova|Scaffold| MOON| FedProto| FedBN |FedFA |FedGLCL |
> |:-----:|:----:|:-----:|:----:|:-----:|:----:|:-----:|:----:|:-----:|:----:|
> |Office-Caltech-10| 64.99 | 62.46 | 65.50 | 61.33 | 66.78 | 69.37 | 71.87 |74.91| 81.81 |
>
> **References**
>
> [1] Fang, Alex, et al. "Data determines distributional robustness in contrastive language image pre-training (clip)." International Conference on Machine Learning. PMLR, 2022.

---

### Official Review · Reviewer_yM3c · 2024-11-04

**Soundness:** 3
**Presentation:** 2
**Contribution:** 2
**Rating:** 6
**Confidence:** 2

**Summary:**

The paper introduces FedGLCL, a FL framework designed to improve the handling of non-IID data in FL environments. The authors leverage language-driven representation learning (e.g. CLIP, as opposed to label driven), incorporating a pre-trained text encoder to stabilize feature representations. This approach enables the alignment of local image features with a global text feature space, reducing feature disparity across clients. Experiments demonstrate that FedGLCL outperforms traditional label-driven FL methods in various non-IID scenarios, including label and feature distribution skew.

**Strengths:**

1. The proposed method is novel - mitigating non-iid data issue in FL on image classification pre-training with text encoders.

2. The experiment results are good, the authors experimented on many datasets, comparing against many baselines, highlighting the generalizability of their method.

3. The author provided theoretical guarantees to the generalizability of the proposed method.

**Weaknesses:**

1. The use of image-text pre-training is prevalent, and the non-iid data issue is not limited to FL settings, so adding text encoders to mitigate the non-iid issue shouldn't be new. Honestly I am not familiar with the FL literature, so I don't know how much this would add to the FL community.

2. The experiments are somewhat limited to small-scaled datasets. Performing experiments on larger scaled datasets (e.g. Imagenet) would make the claims stronger.

**Questions:**

In this paper, federated learning is applied to a specific type of image-text pre-training. I suggest to make this clear in the abstract and throughout the paper - as there are also other types of federated learning as well (e.g. text only)

158 - 161: Can you explain more how your work differs from other works that uses CLIP in FL training (e.g. Lu et al., 2023, Guo et al., 2023, Shi et al., 2023)? In particular - what is "the philosophy of language-driven representation learning"?

---

> ### Author Response · Authors · 2024-11-20
> **Official Comment by Authors**
>
> We thank the reviewer for his/her constructive comments and provide our point-wise replies as follows.
>
> > **Q1:** Contribution to the FL community.
>
> We clarify the contribution of our work as follows.
>
> (1) **First attempt in FL:** To the best of our knowledge, FedGLCL is the first attempt to introduce language-driven representation learning into FL. Although language-driven representation learning is used in centralized settings [1, 2], we address an entirely different problem i.e., non-IID data in the FL setting.
>
> (2) **Label-driven vs. language-driven:** Existing FL methods typically adopt the label-driven training paradigm, leading to significant drift in local models  when handling non-IID data. In this work, we found that using a global text embedding for supervision can effectively mitigate feature differences among local models and prevent overfitting, thus alleviating the non-IID issues in FL. This provides more insights and opens up a new prospect for addressing this challenge.
>
> (3) **Theoretical and empirical analyses:** Furthermore, we offer  theoretical and empirical analyses to gain a deeper understanding of our approach. Extensive experiments show that our method consistently outperforms various state-of-the-art approaches across two different non-IID scenarios, demonstrating  its effectiveness.
>
>
>
> > **Q2:** Add larger-scale datasets.
>
> The four datasets used in the submitted paper are popular benchmarks in heterogeneous FL, which ensures a fair comparison with previous methods. To the best of our knowledge, in heterogeneous FL, no results have been reported on ImageNet, making it difficult to compare with competing methods. Following [3], we conduct experiments using another larger-scale dataset, Tiny-ImageNet [4], which has a similar data distribution to ImageNet. Detailed information about Tiny-ImageNet is provided below, which has been included in Table 5.
>
> |Property |Tiny-ImageNet|
> |:-----|:----:|
> |# of  train samples| 100000 |
> |# of  test samples| 10000 |
> |# of classes| 200 |
> |Image size|(64, 64, 3) |
>
> We use the same experimental setup as for CIFAR-100, and $\beta$ is set to 0.5. The results are presented below. As shown, FedGLCL consistently achieves the best performance compared to all competing methods, further demonstrating the effectiveness of our method. The results have been included in the revision (Appendix D.7 Table 16).
>
> |Method |FedAvg|FedProx |FedNova|Scaffold| MOON| FedProto| FedLC |FedConcat |FedGLCL |
> |:----:|:----:|:-----:|:----:|:-----:|:----:|:-----:|:----:|:-----:|:----:|
> |Tiny-ImageNet| 47.54 |46.61 | 47.11 | 48.09 | 48.97  | 48.16 | 45.97 |49.24|51.05 |
>
>
>
>
> > **Q3:**  Emphasize the type of FedGLCL.
>
> Our work is not a type of pre-training. Instead, it leverages textual information to conduct supervised learning on specific visual recognition tasks. Therefore, it is a type of image-text learning. To distinguish it from text-only or image-only, we emphasize it in the abstract accordingly:
>
> Line 017-018:  "a novel language-driven FL framework that" to "a novel language-driven FL framework for image-text learning that"
>
>
>
> > **Q4:** Difference with Lu et al., 2023, Guo et al., 2023, Shi et al., 2023.
>
> FedGLCL is fundamentally different from them, we explain the differences as follows.
>
> (1) They directly apply the well-trained CLIP model (including both the image and text encoders) as the core components for their tasks. Therefore, their methods are **highly dependent on** the CLIP model. Instead, FedGLCL does not require any modules from CLIP as it can use arbitrary language models as the global text encoder.
>
> (2) They use labels for supervision, which remains the traditional **label-driven** FL approach. FedGLCL is a **language-driven** FL method that aligns the local image feature space with the global text feature space through global text supervision.
>
> (3) Since they use image encoders of CLIP, they are easy to incur domain shift in visual tasks. In contrast, FedGLCL is robust to domain shift as it learns the visual encoder from scratch. For instance, the word "cat" holds the same semantic in both natural images and cartoon image tasks. However, for image backbones, the natural images and the cartoon images represent two entirely different data distributions.

---

> ### Author Response · Authors · 2024-11-20
> **Official Comment by Authors**
>
> > **Q5:** what is "the philosophy of language-driven representation learning"?
>
> Sorry for the confusion. It refers to the conceptual approach or theoretical framework that emphasizes the use of language as supervision in learning meaningful, transferable representations of data. ''the philosophy of'' may cause confusion, so we have removed it in the revision (Line 160).
>
> **References**
>
> [1] Jia, Chao, et al. "Scaling up visual and vision-language representation learning with noisy text supervision." International conference on machine learning. PMLR, 2021.
>
> [2] Radford, Alec, et al. "Learning transferable visual models from natural language supervision." International conference on machine learning. PMLR, 2021.
>
> [3] Li, Qinbin, Bingsheng He, and Dawn Song. "Model-contrastive federated learning." Proceedings of the IEEE/CVF conference on computer vision and pattern recognition. 2021.
>
> [4] https://www.kaggle.com/c/tiny-imagenet/overview.

---

> > ### Author Response · Authors · 2024-11-24
> > **Official Comment by Authors**
> >
> > As the discussion period draws to a close soon, we extend our sincere gratitude to you for the valuable time and insightful comments.
> >
> > We understand that you mentioned, "*I am not familiar with the FL literature*", which may have raised some concerns. We sincerely hope our responses have effectively addressed them.
> >
> > If you have any remaining questions or require further clarification, please do not hesitate to let us know, and we would be glad to provide further explanations
> >
> > Thank you again for your efforts in reviewing our work.

---

> > > ### Comment · Reviewer_yM3c · 2024-11-24
> > > **Reviewer Response**
> > >
> > > I think the authors have addressed my concerns. Therefore, I am raising my score from 5 to 6.

---

> > > > ### Author Response · Authors · 2024-11-24
> > > >
> > > > Thanks for your valuable time to respond to our feedback. We are very happy to see that your concerns have been fully addressed :)

---

### Author Response · Authors · 2024-11-20
**Official Comment by Authors**

Dear reviewers and meta-reviewers,

We appreciate all reviewers for their valuable comments and suggestions. We've revised our manuscript based on reviewers' comments as follows:

(1) For Reviewer yM3c, we have added the results for Tiny-ImageNet in Table 16, with detailed information about the dataset provided in Table 5.

(2) For Reviewer yM3c, we have revised the abstract (Line 017-018) to clarify the type of FedGLCL.

(3) For Reviewer yM3c, we have revised our writting in Line 160.

(4) For Reviewer bW7e, we have discussed the impact of language supervision in Appendix D.2 (Line 1069-1079).

(5) For Reviewer bW7e, we have reported the standard errors for results in Table 17.

(6) For Reviewer bW7e and Q1wc, we have added the results of the experiment using pre-trained image models and a new baseline, FedPCL, in Table 18.

(7) For Reviewer Q1wc, we have added the results on medical image segmentation in Table 19.

(8) For Reviewer Q1wc, we have added the results of our method with Glove in Table  13.

The changes have been highlighted in **blue** in the revised paper. Please see below for our responses to each reviewer. If you have any further questions or suggestions, please feel free to share them on OpenReview.

---

### Meta-Review · Area_Chair_a9vB · 2024-12-20

**Metareview:**

This paper introduces FedGLCL, a language-driven federated learning (FL) framework that integrates global language and local image features via contrastive learning to address challenges posed by non-IID data. By harmonizing local image features with a pre-trained text encoder, FedGLCL ensures uniform feature learning, reduces variance in local representations, and mitigates overfitting to majority classes. Experiments demonstrate its superior performance over state-of-the-art FL algorithms in diverse non-IID settings.

This is a borderline paper. The reviewers have brought up a few issues with this paper, though I think the authors have addressed many of the issues. In my opinion, it would be good to see this paper in ICLR. I would encourage the authors to go through the reviews carefully and address it in the next version.

**Additional Comments On Reviewer Discussion:**

This is a borderline paper. The reviewers have brought up a few issues with this paper, though I think the authors have addressed many of the issues. In my opinion, it would be good to see this paper in ICLR. I would encourage the authors to go through the reviews carefully and address it in the next version.

---

### Decision · Program_Chairs · 2025-01-22

Accept (Poster)